# Localized, High-resolution Geographic Representations with Slepian Functions

**Arjun Rao**[1]   **Ruth Crasto**[2]   **Tessa Ooms**[3]   **David Rolnick**[4 5]   **Konstantin Klemmer**[6 7]   **Marc Rußwurm**[8 3]

## Abstract

Geographic data is fundamentally local. Disease outbreaks cluster in population centers, ecological patterns emerge along coastlines, and economic activity concentrates within country borders. Machine learning models that encode geographic location, however, distribute representational capacity uniformly across the globe, struggling at the fine-grained resolutions that localized applications require. We propose a geographic location encoder built from spherical Slepian functions that concentrate representational capacity inside a region-of-interest and scale to high resolutions without extensive computational demands. For settings requiring global context, we present a hybrid Slepian-Spherical Harmonic encoder that efficiently bridges the tradeoff between local-global performance, while retaining desirable properties such as pole-safety and spherical-surface-distance preservation. Across five tasks spanning classification, regression, and image-augmented prediction, Slepian encodings outperform baselines and retain performance advantages across a wide range of neural network architectures. Our code is available at https://github.com/arjunarao619/SlepianPosEnc.

## 1. Introduction

Learnable functions of geographic coordinates are useful in a variety of critical applications of machine learning such as air quality forecasting (Karimzadeh et al., 2025), crop yield estimation (Tseng et al., 2025b), and species distribution modeling (Mac Aodha et al., 2019). These functions, known as *geographic location encoders*, are commonly modeled as the composition of a neural network wrapped around a non-

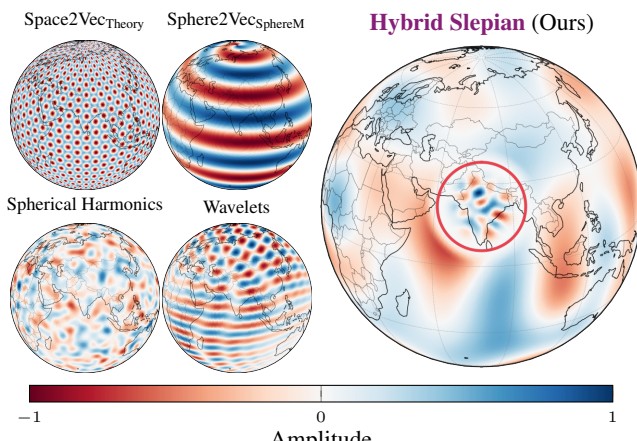

Figure 1. **Slepian functions concentrate a band-limited basis inside a chosen region.** Traditional geographic representations (left) distribute a fixed resolution budget uniformly, forcing a trade-off between global smoothness and local detail. Our proposed hybrid Slepian encoder (right) concentrates high-frequency basis functions exclusively within a region-of-interest (red circle) while preserving global context outside.

parametric positional encoder (Mai et al., 2022; Klemmer et al., 2025b).

Many choices of positional encoder have been studied in recent years. Subsets of the Double Fourier Sphere (DFS) basis, defined on the rectangular plane, have been proven effective and are efficient to compute, but suffer from discontinuity at the poles (Mai et al., 2023; Rußwurm et al., 2024). Spherical harmonics have been proposed as an alternative: these orthogonal functions are the analogue of the Fourier basis but natively defined on the sphere (Rußwurm et al., 2024). Notably, spherical harmonics are supported on the entire globe, making them highly suitable for learning global-scale representations. For example, SatCLIP (Klemmer et al., 2025a) is a general-purpose, pre-trained location encoder that uses spherical harmonics for its positional encoding, resulting in global and pole-safe representations that are useful across a broad range of tasks. Pole-safety refers to the criteria that the encoding should be well-defined everywhere on the sphere, including at the poles where longitude is not meaningful.

Geospatial prediction tasks are often localized — whether it is along coastlines or within administrative boundaries — and many of these tasks require representations that can

---

[1]Department of Computer Science, University of Colorado Boulder [2]Microsoft [3]University of Wageningen, Netherlands [4]McGill University [5]Mila–Quebec AI Institute [6]LGND AI, Inc [7]University College London [8]University of Bonn, Germany. Correspondence to: Arjun Rao <raoarjun@colorado.edu>.

*Proceedings of the 43rd International Conference on Machine Learning*, Seoul, South Korea. PMLR 306, 2026. Copyright 2026 by the author(s).

capture fine-grained patterns at higher spatial resolutions (Rolf et al., 2024). However, the spatial resolution achievable using spherical harmonics globally is limited due to numerical instability, and global resolutions used in practice are often insufficient to perform well on localized prediction tasks. For example, Klemmer et al. (2025a) observed that SatCLIP performs poorly on the California Housing dataset (Pace & Barry, 1997), where the task is to predict median house prices within housing districts in California. Ultimately, positional encoders that distribute spatial resolution uniformly across the globe are ill-suited to the localized nature of many geospatial prediction tasks.

To address these challenges, **we propose to use Slepian functions as a positional encoder basis for achieving high-resolution, spatially-concentrated, and computationally feasible representations of geographic location**. We further introduce a hybrid Slepian-Spherical Harmonics (SH) positional encoder that is capable of scaling to high resolutions locally while retaining relevant coarser, global context. Finally, we extend our proposed positional encoder to cover multiple localized regions on Earth, and demonstrate that a one-dimensional generalization of our Slepian encoder can model temporal data. Our main findings are:

1. Slepian encoders concentrate geographic *representational capacity* within a spatial (or temporal) region of interest (ROI), leading to improved downstream prediction performance over global models across five different tasks.

2. Our hybrid Slepian-SH encoder for localized tasks with global context bridges the unfavorable local-global tradeoff in geospatial machine learning.

3. Slepian encoders are computationally feasible and more memory efficient compared to SH encoders.

## 2. Related Work

**Geographic Location Encoders** learn continuous functions mapping geographic coordinates (i.e., longitude, latitude) on $\mathbb{S}^2$ to target values. This is typically done with a positional encoding (PE) projecting coordinates into a high-dimensional feature space followed by a neural network that projects these features into the final output space (Mai et al., 2022). A central challenge for location encoders is representing high-frequency spatial information: neural networks exhibit spectral bias toward low frequencies (Rahaman et al., 2019), motivating Fourier feature mappings (Tancik et al., 2020) and periodic activations (Sitzmann et al., 2020) as solutions. Popular geographic location encoders include multi-scale sinusoidal functions inspired by grid cells (Mai et al., 2020; 2023), Random Fourier Features (Cepeda et al., 2023), and spherical harmonics providing an orthogonal, sphere-native basis (Rußwurm et al., 2024). Spatial resolu-

tion is controlled by hyperparameters: the number and range of frequency scales, or for spherical harmonics, the maximum degree $L$ yielding resolution approximately $20{,}000/L$ km (Wang et al., 2025). Contrastive image-location matching has emerged as a dominant location encoder pretraining paradigm, with SatCLIP (Klemmer et al., 2025a) and GeoCLIP (Cepeda et al., 2023) demonstrating strong transfer to downstream tasks. Recent hybrid approaches combine spherical harmonics with image-augmented retrieval (Dhakal et al., 2025) or pair implicit and explicit representations for species distribution modeling (Yuan & Zhao, 2024). These approaches, however, share a fundamental limitation: higher resolutions capture finer spatial frequencies but remain distributed across the entire sphere, yielding dense representations that scale poorly and lack regional focus.

**Slepian Functions** have been utilized as natural solutions to modeling natural phenomena in smaller regions. Slepians, also known as prolate spheroidal wave functions (PSWFs), are solutions to the concentration problem of finding band-limited functions maximally concentrated within a finite spatial (or temporal) interval (Slepian & Pollak, 1961). Landau & Pollak (1962) subsequently established fundamental limits on simultaneous time-frequency concentration, and Slepian (1964) extended the framework to multiple dimensions and discrete sequences (Slepian, 1978). The discrete formulation became widely adopted through Thomson (1982)'s multitaper spectral estimation method, which remains standard in signal processing and neuroscience applications (Prerau et al., 2017). On the sphere, Simons et al. (2006) constructed band-limited functions optimally concentrated within arbitrary regions, enabling localized analysis in geophysics and planetary science. Spherical Slepians have been adopted in satellite geodesy, including GRACE-based studies of ice mass change in Greenland, Antarctica, and smaller glaciated regions (Harig & Simons, 2012; 2015; 2016; Von Hippel & Harig, 2019), earthquake-related gravity changes (Han & Simons, 2008; Wang et al., 2012), and terrestrial water storage variability (Rateb et al., 2017). In contrast to prior work that uses spherical Slepians to analyze or invert measured geophysical signals, we use them as geographic position encoders to learn localized coordinate representations.

## 3. Method

Let $x = (\lambda, \phi) \in \mathbb{S}^2$ denote a point on the unit sphere, parameterized by longitude $\lambda \in [-\pi, \pi)$ and latitude $\phi \in [-\pi/2, \pi/2]$. A geographic location encoder is a function $\Phi : \mathbb{S}^2 \to \mathbb{R}^d$ of the form $f(x) = \mathrm{NN}(\Phi(x))$, where $\Phi : \mathbb{S}^2 \to \mathbb{R}^D$ is a positional encoder and $\mathrm{NN} : \mathbb{R}^D \to \mathbb{R}^d$ is a neural network (Rußwurm et al., 2024; Klemmer et al., 2025a; Mai et al., 2022).

## 3.1. Spherical Harmonics (SH) Positional Encoder

The canonical basis for $\mathbb{S}^2$, and a common choice for $\Phi$ is the set of real spherical harmonics $\{Y_\ell^m\}$ since they are well-defined on $\mathbb{S}^2$ and form an orthonormal basis. They are the spherical analogue of the Fourier basis (Rußwurm et al., 2024). Here, $\ell \geq 0$ is the harmonic degree (spatial frequency) and $m$ is the order, with $-\ell \leq m \leq \ell$. In practice, we work with a band-limited subspace up to maximum degree $L$:

$$\mathcal{H}_L = \text{span}\{Y_\ell^m : 0 \leq \ell \leq L, -\ell \leq m \leq \ell\},$$

which has dimension $D_L := (L+1)^2$. The SH positional encoder in Rußwurm et al. (2024) evaluates all basis functions at a point $x \in \mathbb{S}^2$:

$$\Phi_{\text{SH}}(x) = [Y_\ell^m(x) : 0 \leq \ell \leq L, -\ell \leq m \leq \ell] \in \mathbb{R}^{D_L}.$$

This basis is global: each $Y_\ell^m$ has support on the entire sphere. **To resolve fine local detail, one must raise $L$, which increases feature dimension as $D_L = (L+1)^2$ and yields a dense, costly representation.**

**Numerical Instability of SH.** Another practical barrier to high-resolution location encoding using SH is numerical instability. The normalization constant for $Y_\ell^m$ is

$$N_{\ell m}^2 = \frac{2\ell+1}{4\pi} \frac{(\ell-|m|)!}{(\ell+|m|)!},$$

which decays rapidly as $N_{\ell m} \sim \ell^{-m}$ for $m > 0$ (Conrad, 2016). In contrast, the standard recurrence for associated Legendre polynomials,

$$(\ell-m+1) P_{\ell+1}^m(t) = (2\ell+1)\, t\, P_\ell^m(t) - (\ell+m) P_{\ell-1}^m(t),$$

propagates coefficients that grow with $\ell$. Multiplying large intermediate values by a vanishing normalization is numerically ill-conditioned, producing non-finite values in finite-precision arithmetic (Wieczorek & Meschede, 2018; Holmes & Featherstone, 2002). In 32-bit precision, this occurs beyond modest bandlimits ($L \gtrsim 40$) (Rußwurm et al., 2024). The problem is exacerbated on modern accelerators, where mixed-precision training (FP16/BF16) is standard (Micikevicius et al., 2018), further lowering the stability threshold. **This numerical barrier prevents SH from reaching the high spatial resolutions required by localized tasks.**

## 3.2. Slepian-Based Positional Encoder

We extend the global SH positional encoding to a *regional*, spatio-spectrally concentrated basis. Given a closed region $R \subset \mathbb{S}^2$, the Slepian concentration problem seeks band-limited functions $h \in \mathcal{H}_{L_r}$ that maximize the energy ratio

$$\mu = \frac{\int_R |h(x)|^2 \, ds(x)}{\int_{\mathbb{S}^2} |h(x)|^2 \, ds(x)} \in [0,1), \quad (1)$$

where $L_r$ is a high-resolution regional bandlimit. This variational problem is solved in the spectral domain by finding the eigenvectors of a $D_{L_r} \times D_{L_r}$ symmetric concentration matrix $\mathbf{K}$:

$$\mathbf{Kh} = \mu\mathbf{h}, \quad K_{\ell m, \ell' m'} = \int_R Y_\ell^m(x)\, Y_{\ell'}^{m'}(x)\, ds(x). \quad (2)$$

This yields $D_{L_r}$ eigenfunctions $\{g_n\}_{n=1}^{D_{L_r}}$ (Slepian functions), which are orthonormal on $\mathbb{S}^2$ and mutually orthogonal over $R$. Their eigenvalues $\{\mu_n\}_{n=1}^{D_{L_r}}$, ordered $\mu_1 \geq \cdots \geq \mu_{D_{L_r}}$, quantify each function's energy concentration inside $R$.

A key property of this spectrum is the regional **Shannon number**, $N(R, L_r)$, defined as the trace of the concentration matrix:

$$N(R, L_r) = \text{tr}(\mathbf{K}) = \sum_{n=1}^{D_{L_r}} \mu_n \approx \frac{\text{area}(R)}{4\pi} (L_r+1)^2, \quad (3)$$

where $\text{area}(R)$ is the surface area of $R$ on the unit sphere. As shown in Figure 2, the spectrum $\{\mu_n\}$ exhibits a sharp transition: $\mu_n \approx 1$ for $n \lesssim N(R, L_r)$ and $\mu_n \approx 0$ thereafter. For geographic location encoding, $N(R, L_r)$ serves as the intrinsic dimensionality or "information-theoretic budget" available *inside* $R$ at bandlimit $L_r$. Motivated by prior work showing that the intrinsic dimensionality of geographic INRs reflects the true information content encoded (Rao et al., 2026), we select the first $K = \lceil N(R, L_r) \rceil$ well-concentrated eigenfunctions to form our regional positional encoder:

$$\Phi_{\text{Slep}}(x) = [g_1(x), g_2(x), \ldots, g_K(x)] \in \mathbb{R}^K. \quad (4)$$

The eigenvalue ordering induces a natural coarse-to-fine hierarchy, with lower-indexed modes capturing broad regional trends and higher-indexed modes resolving finer spatial detail. Beyond numerical stability, Slepians provide a memory-efficient representation. The regional encoder dimension $K = \lceil N(R, L_r) \rceil$ scales with the area fraction $f_R = \text{area}(R)/4\pi$, not the full sphere. For Sri Lanka ($f_R \approx 1.29 \times 10^{-4}$), bandlimit $L_r = 256$ yields only $K \approx 9$ regional modes versus $D_{L_r} \approx 6.6 \times 10^4$ for a global basis. This sparsity is intrinsic: for small regions, most SH energy lies outside $R$, so few modes concentrate within it.

**Spherical Slepian Caps.** For an arbitrary region $R$, computing Slepian functions requires solving a $D_{L_r} \times D_{L_r}$ eigenproblem, which becomes prohibitive at high bandlimits. To make high-resolution regional Slepians practical, we build our regional basis from **spherical Slepian caps** (Bates et al., 2017). A spherical cap is a simple region on the sphere defined by taking all points within an angular radius $\Theta$ of a chosen center (i.e., a circular "patch" on Earth). For caps, the Slepian concentration matrix block-diagonalizes

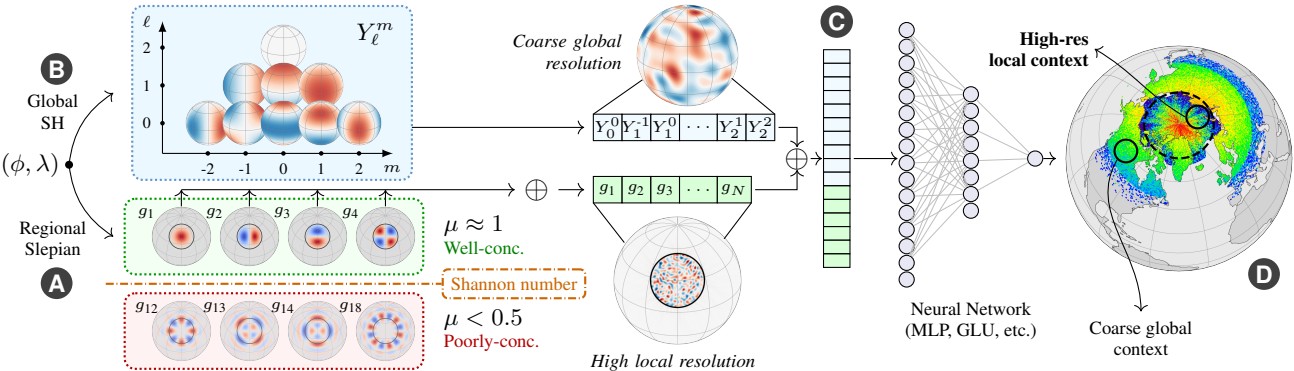

*Figure 2.* **Constructing our hybrid Slepian encoder.** ( **A** ) For a region encompassing the geographic coordinate, we compute Slepian eigenfunctions $\{g_n\}$ ordered by their eigenvalues $\mu_n$ and discard modes with eigenvalues under the regional Shannon number to form our Slepian positional encoder. These modes are concatenated with global SH basis functions ( **B** ) of a coarse resolution to form our hybrid positional embedding ( **C** ). The hybrid positional embedding is passed to a neural network to form our final location encoder that captures high-resolution detail within the region-of-interest (ROI) while retaining global context ( **D** ).

by order $m$, reducing the eigenproblem in Equation (2) to independent blocks of size at most $L_r \times L_r$. The number of well-concentrated modes is explicit:

$$N_\Theta(L_r) \;=\; \frac{1 - \cos\Theta}{2}\,(L_r + 1)^2,$$

so $\Theta$ directly controls the local information budget. In practice, we compute cap Slepians once at high $L_r$, rotate them to the desired center, and retain the top $N_\Theta(L_r)$ modes. This avoids constructing the dense matrix $\mathbf{K}$ in the full SH space and makes high-$L_r$ encoders feasible. We visualize our hybrid encoder with a high-resolution cap in Figure 1 and compare cap versus arbitrary-region computation in appendix Figure 12.

### 3.3. Hybrid Slepian-SH Positional Encoder

Our Slepian basis $\Phi_{\text{Slep}}(x)$ captures high-frequency detail *inside* $R$, but has negligible energy outside of it. Using it alone sacrifices global context relevant to many geospatial tasks (Cole et al., 2023; Mac Aodha et al., 2019). We therefore propose a **hybrid encoder** that operates at two bandlimits: a high-resolution regional bandlimit $L_r$ and a coarse global bandlimit $L_g \ll L_r$. The encoder concatenates two components:

1. **Regional Slepian Component:** The basis $\Phi_{\text{Slep}}(x)$ as defined in Equation (4), computed at bandlimit $L_r$.
2. **Global SH Component:** A standard SH basis $\Phi_{\text{SH}}(x)$ computed at bandlimit $L_g$.

Our hybrid positional encoder is the concatenation of these two bases:

$$\Phi_{\text{Hybrid}}(x) \;=\; \text{Concat}\big[\Phi_{\text{Slep}}(x),\, \Phi_{\text{SH}}(x)\big]. \quad (5)$$

This construction extends naturally to multiple regions: given $C$ regions, we concatenate their encoders

$\{\Phi_{\text{Slep}}^{(c)}(x)\}_{c=1}^{C}$ with the global SH component. Since each Slepian basis concentrates energy inside its region, cross-region interference is negligible. We demonstrate a multi-region application in Section A.1.

**Slepian Functions are pole-safe, and reduce to global SH when computed over $\mathbb{S}^2$.** Intuitively, moving $\lambda$ while keeping $\phi = \pm\frac{\pi}{2}$ should not change the encoding. Similarly, the geometry induced by $\Phi$ should respect spherical surface distance, meaning that points that are close (far) in great-circle distance on $\mathbb{S}^2$ should remain close (far) in embedding space, avoiding distortions of local neighborhoods and long-range relationships. Our Slepian-based encoder satisfies both these critical modeling criteria since it is a principled generalization of the global SH basis. This is evident in the degenerate case where $R = \mathbb{S}^2$. By orthonormality of spherical harmonics, the concentration matrix $\mathbf{K}$ becomes the identity:

$$K_{\ell m,\,\ell'm'} \;=\; \int_{\mathbb{S}^2} Y_\ell^m(x)\, Y_{\ell'}^{m'}(x)\, ds(x) \;=\; \delta_{\ell\ell'}\,\delta_{mm'}.$$

The eigenproblem $\mathbf{Kh} = \mu\mathbf{h}$ thus has all eigenvalues $\mu = 1$, and the Slepian functions $\{g_n\}$ reduce to the global SH basis $\{Y_\ell^m\}$.

More critically for modeling in polar regions, Slepian functions are inherently pole-safe. Each $g_n$ is a finite linear combination of spherical harmonics:

$$g_n(x) \;=\; \sum_{\ell=0}^{L_r} \sum_{m=-\ell}^{\ell} h_{\ell m}^{(n)}\, Y_\ell^m(x).$$

Since $Y_\ell^m$ are analytic across the entire sphere, including at the poles (Khalid et al., 2016), $g_n$ inherits this regularity. This regularity is a practical advantage for high-resolution regional encoding. Unlike spherical wavelets based on stereographic projection (Cai & Balestriero, 2025), which break

down at the poles, our Slepian-SH encoder provides high-resolution local features while remaining sphere-native.[1]

**Extension to Spatio-Temporal Data.** The concentration principle underlying our spatial Slepian positional encoder in Equation (4) extends naturally to the *temporal* domain. As noted earlier, spatial Slepians solve the problem of concentrating band-limited functions inside a region $R \in \mathbb{S}^2$. The temporal analogue, Discrete Prolate Spheroidal Sequences (DPSS) (Slepian, 1978), solves the dual problem: given a sequence of finite duration $N_t$ timesteps, find orthonormal sequences that optimally concentrate their energy inside a target frequency band. In temporal modeling, lower frequencies correspond to slow-varying patterns such as seasonal cycles, while higher frequencies capture rapid fluctuations; the half-bandwidth $W$ thus controls which temporal scales the encoder emphasizes. Analogous to the spatial Shannon number in Equation (3), the temporal Shannon number $k_t \approx 2N_tW$ governs how many sequences concentrate well. In our temporal Slepian we retain the leading $K_t$ well-concentrated sequences and combine them with a global SH spatial encoder via concatenation: $\Phi_{\mathrm{ST}}(x,t) = \mathrm{Concat}[\Phi_{\mathrm{SH}}(x), \Phi_{\mathrm{Time}}(t)]$. This space-time construction mirrors the joint location–time encoder of Mickisch et al. (2025), where we replace the Fourier-based temporal component with our spectrally-concentrated DPSS basis. Full details are provided in Appendix A.4.

## 4. Experiments

In this section, we describe the two broad sets of tasks we use to evaluate our hybrid Slepian-SH encoder, and the baseline positional encoders we compare against.

### 4.1. Datasets and Tasks

**Geographic Classification and Regression.** We evaluate on a diverse selection of classification and regression tasks. **(i)** California Housing (Pace & Barry, 1997) is a classic spatial regression dataset where the task is to predict median house prices from geographic coordinates and other features. **(ii)** Japan Prefectures is a classification task we construct over geographic coordinates similar to the global country classification benchmark in Klemmer et al. (2025a). In this task, the target for classification is one of 47 prefectures in Japan: small, densely packed administrative regions. Test splits only include points that are within 2 km of a prefecture boundary, creating a challenging classification task. **(iii)** To evaluate pole-safety, we use an oceanographic interpolation task based on the synthetic mean sea surface (MSS) dataset introduced by Chen et al. (2025). The target variable is a static MSS over the Arctic, constructed from multi year

CryoSat-2 sea surface height measurements and gridded on a 5 km grid. **(iv)** To test our 1-D Slepian encoder's spatio-temporal modeling ability, we use the AI2 Climate Emulator (ACE) dataset (Watt-Meyer et al., 2023), which contains one year of simulated atmospheric data at 6-hourly resolution. The task is to predict air temperature at 8 pressure levels given a spatio-temporal coordinate $(\lambda, \phi, t)$.

**Geo-Aware Image Prediction.** Fusing location embeddings with image features has been shown to improve the performance of image understanding models and has become a standard way of benchmarking location encoders (Klemmer et al., 2025a; Wu et al., 2024). We evaluate two geo-aware prediction tasks using frozen image encoders: building density regression, where location embeddings are fused via concatenation with the image embeddings, and presence-only species distribution modeling, where location embeddings are combined with image features by element-wise multiplication. For building density regression, we use the OpenBuildings dataset of high-resolution satellite imagery (Sirko et al., 2021), evaluating on four regions (South Florida, Dhaka, Mexico City, Maharashtra) with two choices of multi-modal encoder (AlphaEarth and Galileo). To test performance across different spatial resolutions, we apply Gaussian smoothing to the ground-truth target using kernel values ($\sigma$) ranging from 0 to 40 km. At $\sigma = 0$, building density is a noisy, high-resolution target with little spatial autocorrelation, while smoothing creates targets that roughly represent density at a coarser levels.

For species distribution modeling, we follow the SINR framework (Cole et al., 2023), training location encoders on presence-only observations from iNaturalist (Van Horn et al., 2018) with globally-sampled pseudo-negatives: random background locations treated as absences during training to compensate for the lack of true negative labels (Cole et al., 2023). At inference, location predictions serve as a spatial prior to re-weight outputs from a frozen Xception classifier (Chollet, 2017) via element-wise multiplication. Our hybrid encoder uses a global SH backbone ($L_{\mathrm{g}} = 10$) combined with two Slepian caps: one centered over the continental United States (25° radius) and one over Europe (20°). This multi-cap configuration specifically tests the hybrid design: purely regional encoders fail because pseudo-negatives are sampled globally and locations outside caps receive negligible representational capacity, while purely global encoders lack spatial resolution for fine-scale species boundaries. We evaluate on the eBird Status and trends (S&T) and The International Union for Conservation of Nature's (IUCN) expert species range maps (Fink et al., 2020; List, 2004). The S&T benchmark consists of 535 bird species with a focus on North America. The IUCN benchmark is a critical test of out-of-cap generalization since it includes species ranges far beyond the cap regions.

---

[1]Sphere-nativity and pole-safety were introduced as desiderata for geographic location encoders in Mai et al. (2023).

## 4.2. Model Details

**Baseline Positional Encoders.** We compare our hybrid Slepian-SH encoder against a comprehensive suite of baseline positional encoders. These include simple transformations of lat/lon such as Direct (normalized coordinates), Cartesian3D (3D unit sphere embedding) (Tseng et al., 2023), and Wrap (a 4-dimensional sine-cosine encoding) (Mac Aodha et al., 2019); multi-frequency Fourier encodings such as Grid and Theory (Mai et al., 2020), Double-Fourier Sphere-based encodings Sphere$^{\{C,M,C+,M+\}}$ (Mai et al., 2023), and spherical wavelets from Cai & Balestriero (2025). We include two Random Fourier Feature (RFF) variants: Planar RFF (Tancik et al., 2020) uses a single-layer random feature expansion with frozen Matérn-derived frequencies, while Deep RFF (Chen et al., 2025) stacks multiple RFF layers with skip connections and learnable output projections. These baselines are described in more detail in Appendix B.1.

**Neural Network Architectures.** The output of the non-parametric position encoder is passed to a trainable neural network. For geographic classification/regression tasks, we use a 3-layer MLP with ReLU activation and dropout $p = 0.1$. For building density estimation, we directly concatenate positional encodings with multi-modal features, and pass the result through a 2-layer bottleneck network followed by a regression head. More training details and hyperparameter specifications are provided in Appendix B.1. In Appendix Table 5, we evaluate and discuss different neural network architectures in detail.

**Slepian Encoder Implementation** Our Slepian cap computation is implemented using `shtools` (Wieczorek & Meschede, 2018). Our cap center and radius are manually chosen to encompass all splits of the task-dataset within the target region. When making comparisons to global SH encoder, we use the analytic computation of SH coefficients. More details can be found in Appendix B.4.

## 5. Results

**Slepian encodings improve supervised geographic prediction across tasks.** Table 1 shows that the Slepian-based positional encoder improves performance on all supervised prediction tasks spanning regression (California housing), classification (Japan prefectures), and regional interpolation (Arctic MSS). This supports our claim that a spatio-spectrally concentrated basis is a strong inductive bias for localized prediction from geographic coordinates. We note that gains in performance come from spatio-spectral concentration and not simply higher dimensionality and expressivity; encoders with higher ambient dimensions such as planar RFFs and SH ($L{=}40$) achieve much weaker results. Across all three tasks, our best Slepian encoder (Slepian, $L{=}120$)

*Table 1.* **Supervised Geographic Regression and Classification Results.** We compare our Slepian-only and hybrid Slepian-spatial encoder against popular positional encoding baselines on a California housing price regression (Pace & Barry, 1997), Japan prefecture classification, and an Arctic Mean-Sea-Surface (MSS) interpolation task (Chen et al., 2025). Slepian functions produce varied dimensionality of embeddings depending on the cap size used per task, shown with forward-slash-separated values. Results averaged with $1 \times$ SD over 5 random seeds. Best and second-best results are **bolded** and underlined respectively.

| Positional Encoder | Dim. | Cali Housing $R^2 \uparrow$ | Japan Prefectures Acc $\uparrow$ | MSS $R^2 \uparrow$ |
|---|---|---|---|---|
| Direct | 2 | 0.05±0.08 | 0.46±0.03 | 0.84±0.00 |
| Cartesian3D | 3 | 0.37±0.01 | 0.71±0.02 | 0.84±0.00 |
| Wrap | 4 | 0.36±0.01 | 0.70±0.02 | 0.89±0.00 |
| Grid | 64 | 0.45±0.01 | 0.81±0.01 | 0.87±0.01 |
| SphereC | 48 | 0.53±0.00 | 0.83±0.07 | 0.87±0.00 |
| SphereC+ | 112 | 0.44±0.06 | 0.83±0.00 | 0.86±0.01 |
| SphereM | 80 | 0.41±0.03 | 0.82±0.01 | 0.91±0.00 |
| SphereM+ | 144 | 0.41±0.05 | 0.82±0.01 | 0.90±0.00 |
| Space2Vec (Theory) | 96 | 0.41±0.02 | 0.84±0.00 | 0.88±0.02 |
| Planar RFF | 2000 | 0.42±0.00 | 0.59±0.04 | 0.79±0.00 |
| Deep RFF | 64 | 0.57±0.01 | 0.81±0.02 | 0.78±0.00 |
| Wavelets | 256 | 0.34±0.01 | 0.80±0.01 | Diverge |
| SH ($L{=}10$) | 121 | 0.45±0.04 | 0.83±0.01 | 0.86±0.00 |
| SH ($L{=}40$) | 1681 | 0.64±0.01 | 0.88±0.01 | Diverge |
| **Slepian ($L{=}40$)** | 12/23/78 | 0.64±0.01 | 0.86±0.00 | 0.95±0.01 |
| **Slepian ($L{=}80$)** | 44/69/307 | 0.66±0.01 | 0.89±0.01 | 0.97±0.00 |
| **Slepian ($L{=}120$)** | 86/144/685 | 0.69±0.01 | 0.89±0.00 | 0.97±0.00 |
| **Hybrid ($L_r{=}40$)** | 112/123/178 | 0.65±0.01 | 0.88±0.01 | 0.97±0.00 |
| **Hybrid ($L_r{=}80$)** | 144/169/407 | 0.68±0.01 | 0.90±0.01 | 0.98±0.00 |
| **Hybrid ($L_r{=}120$)** | 186/244/785 | **0.71**±0.01 | **0.92**±0.00 | **0.98**±0.00 |

achieves the strongest performance while using only 186 to 785 embedding dimensions compared to 1600 dimensions for high-bandlimit global SH ($L{=}40$), i.e., a $1/10^{\text{th}}$-to-$4/10^{\text{th}}$ smaller embedding. From Figure 3, we also find that Slepian encodings are well-suited for high-resolution geographic interpolation tasks. The Arctic MSS reconstruction task is an evaluation setting where high-resolution structure as well as pole-safety is essential. The advantage of using Slepian encodings for this task is qualitatively visible in Figure 3. Compared to baselines, reconstructions using Slepians better preserve small-scale spatial detail, recovering sharper gradients and localized anomalies while reducing large-scale smoothing and structured artifacts at the poles. We visualize additional baselines on this task in appendix Figure 10. Finally, in Appendix Table 5 we experiment with different choices of neural network architecture, and find that Slepian encodings achieve the top results regardless of neural network choice.

**Slepian encodings concentrate representational capacity *inside* the ROI.** From Figure 4, we observe that Slepian encodings allocate all their representational capacity inside the region-of-interest, with minimal contribution out-of-cap. We test this by systematically varying cap radius to encompass 10–100% of test points. All points are encoded with the same Slepian-SH hybrid basis, but points outside the cap receive attenuated representations since Slepian functions

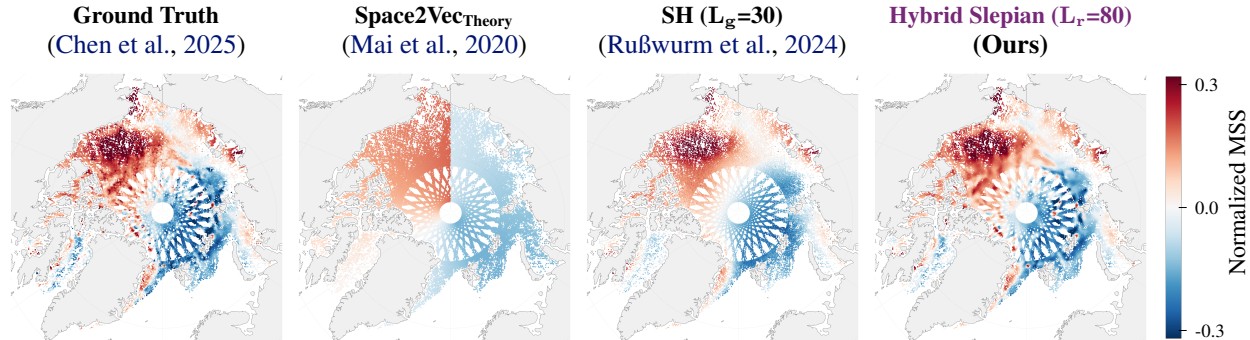

*Figure 3.* **Slepian-based location encoders better preserve fine-scale spatial/geographic detail.** We visualize Arctic Mean Sea Surface (MSS) interpolation results proposed in (Chen et al., 2025). Spherical Harmonics at $L \geq 40$ diverges and does not produce a valid reconstruction. Interpolation results reinforce that Slepians inherit the pole-safe property of SH (Section 3.3) unlike Space2Vec and several DFS-derived location encoders. Line-like artifacts are visible around the poles due to the nature of satellite track measurements. We visualize the interpolation results of additional baselines in Figure 10.

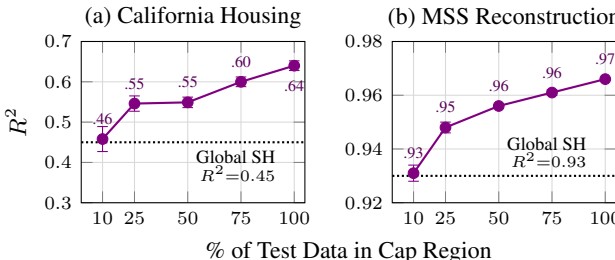

*Figure 4.* **Slepian functions concentrate representational capacity within the cap region.** As cap coverage increases, $R^2$ improves monotonically, while global spherical harmonics (dotted line) remain constant regardless of the spatial extent of interest. Error bars show $1\times$ SD over 5 random seeds.

*Table 2.* **Species distribution modeling results on iNaturalist with SINR.** Mean Average Precision (mAP) on S&T Birds (535 species) and IUCN range maps (2418 species). LinNet is a linear classifier; ResFCNet uses a deep network.

| Encoding | Config | LinNet | | ResFCNet | |
|---|---|---|---|---|---|
| | | S&T ↑ | IUCN ↑ | S&T ↑ | IUCN ↑ |
| Wrap | — | 0.250 | 0.009 | 0.732 | 0.627 |
| SH | $L_g=10$ | 0.637 | 0.342 | **0.736** | 0.696 |
| | $L_g=30$ | 0.560 | 0.102 | 0.716 | 0.691 |
| | $L_g=40$ | 0.440 | 0.069 | 0.673 | 0.645 |
| Slepian ($L_g=0$) | $L_r=10$ | 0.264 | 0.011 | 0.730 | 0.679 |
| | $L_r=20$ | 0.269 | 0.011 | 0.711 | 0.682 |
| | $L_r=30$ | 0.271 | 0.013 | 0.692 | 0.665 |
| | $L_r=40$ | 0.298 | 0.013 | 0.678 | 0.656 |
| **Hybrid Slepian** ($L_g=10$) | $L_r=10$ | 0.695 | 0.433 | 0.711 | **0.704** |
| | $L_r=20$ | **0.704** | 0.460 | 0.711 | 0.701 |
| | $L_r=30$ | 0.696 | 0.478 | 0.691 | 0.702 |
| | $L_r=40$ | 0.672 | **0.479** | 0.665 | 0.694 |

concentrate their energy within the cap. If spectral concentration truly increases representational capacity, average performance should improve as more test points fall inside the cap region. Figure 4 confirms this: $R^2$ increases monotonically with coverage while global SH with a matched embedding dimension remains flat. For the California housing task, we find that just 3 well-concentrated Slepian modes at 25% cap coverage (cap radius = $2.26°$) produces a 19% relative improvement in task performance compared to global SH.

**DPSS temporal encodings improve spatio-temporal prediction.** From Table 3, DPSS encodings achieve mean RMSE of 1.344 on the ACE climate prediction task, outperforming the Fourier baseline (1.477) by 9%. This improvement is consistent across all atmospheric levels from stratosphere to surface (Appendix Table 6), and is robust across a range of bandwidth settings. Full experimental details are in Appendix A.4.

**Slepian encodings improve performance on localized geo-aware image regression.** From Figure 5, we find that (i) location embeddings significantly improve performance on the OpenBuildings dataset across all spatial scales, and

(ii) across all cities, Slepian encodings improve performance on moderately-fine frequency scales ($\sigma \in [3, 15]$ km) compared to high-resolution SH. At $\sigma = 5$ km, Slepian $L = 120$ exceeds SH $L = 40$ by 7 to 14% in $R^2$ depending on region and image encoder. As the prediction target is spatially smoothed ($\sigma \to 40$ km), all location-aware methods converge to $R^2 \approx 0.99$, indicating that coarse-scale structure is easily captured by any reasonable basis. Image embeddings alone (gray dashed) stagnate or degrade with increasing $\sigma$.

**Global context helps local tasks.** From Table 1, we find that our hybrid Slepian encoder outperforms a Slepian-only encoder ($L_g = 0$) on all 3 supervised geographic prediction tasks, even when all the tasks only contain geographic data within a localized region. Notably, the Slepian-only encoder still outperforms most baselines, indicating that regional concentration is the primary driver of performance gains, while global context provides a modest but reliable

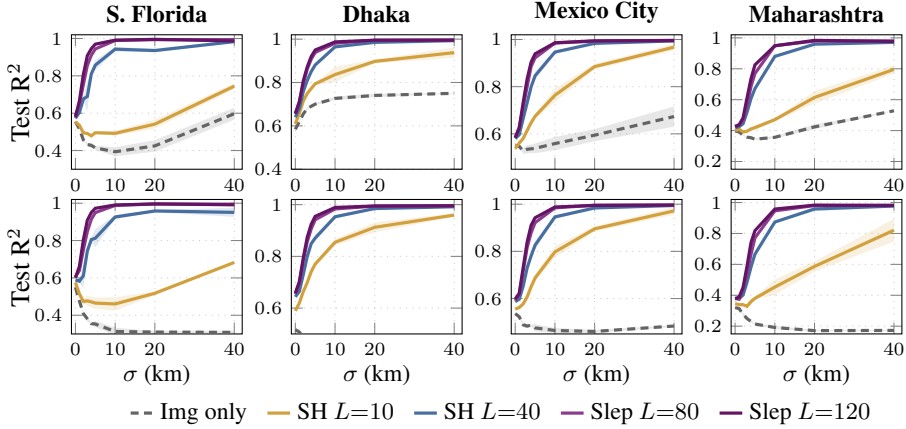

*Figure 5.* **Building density regression** with OpenBuildings (Sirko et al., 2021) using AlphaEarth (top) (Brown et al., 2025) and Galileo (bottom) (Tseng et al., 2025a) image embeddings. Shaded regions indicate $1\pm$ SD over 5 random seeds.

*Table 3.* **Spatio-temporal modeling on the AI2 ACE dataset with 1D Slepians (DPSS).** Mean RMSE (mean $\pm$ std across seeds) averaged across eight pressure levels.

| Model | Mean RMSE |
| --- | --- |
| No Time | $5.703\pm0.001$ |
| Time Copy | $2.428\pm0.018$ |
| Triangle | $3.092\pm0.003$ |
| Monomial | $2.033\pm0.178$ |
| Legendre | $1.532\pm0.285$ |
| Fourier | $1.477\pm0.128$ |
| DPSS ($NW{=}15$) | $\mathbf{1.344\pm0.012}$ |
| DPSS ($NW{=}19$) | $\underline{1.357\pm0.011}$ |

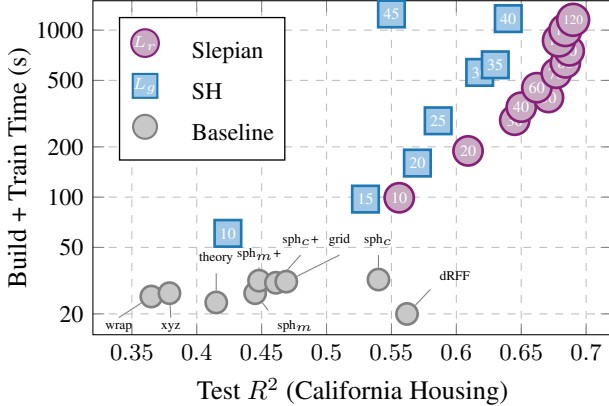

*Figure 6.* **Slepian encoders achieve better performance at a lower computational cost compared to global SH.** Each marker for SH and Slepians corresponds to a different resolution $L$. Slepians consistently appear further right (better $R^2$) on the California housing regression task. At higher resolutions ($L \geq 40$), Spherical Harmonics diverge due to numerical instability. Gray circles show baseline positional encoders, which are fast but underperform.

additional benefit.

**Local context helps global tasks.** The species distribution modeling task introduced for machine learning in Beery et al. (2021) is inherently global. Species ranges and habitats can occupy different continents, but can often cluster in small ecological niches. This presents a unique application of our hybrid encoder due to the need for high local resolutions in addition to global context. From Table 2, we find that performing a presence-only species classification task benefits most with a location prior from our multi-Slepian-cap hybrid encoder, especially when the capacity of the neural network is reduced (linear model). Surprisingly, we observe that our hybrid encoder does well on the IUCN benchmark which contains a large number of species counts in eastern Africa. This constitutes an out-of-cap evaluation setting where high-resolution within-cap context in USA

and Europe aid the hybrid representations out-of-cap. Secondly, we observe that this task does not always benefit from high local resolutions. We provide more detail in Appendix Section A.1. In addition, Slepian functions can aid global tasks by providing high-resolution local context at challenging regions, thereby partially mitigating large-scale smoothing. We demonstrate masked Slepians for global land-ocean classification in Appendix A.2.

**Slepian functions are compute-efficient.** As shown in Figure 6, Slepian encoders dominate the compute-performance Pareto front, surpassing both SH and RFF baselines. While SH is faster at low resolutions ($L = 10$) due to Slepian's initial basis-construction overhead, this cost is rapidly amortized as resolution increases. At matched resolution ($L = 30$), Slepians are both faster and more accurate than SH (289s vs. 558s; $R^2 = 0.645$ vs. 0.618). At matched performance ($R^2 \approx 0.62$), Slepians offer a $3\times$ speedup (189s vs. 558s), and at a matched time budget ($\sim$290s), they achieve significantly higher accuracy ($R^2 = 0.64$ vs. 0.586). Ultimately, the performance gap between Slepian and other encoders continues to widen with increasing resolution. Although other baselines like deep RFFs are fast, their test $R^2$ values are lower compared to both Slepian and SH encoders.

## 6. Discussion and Conclusion

**Limitations.** Our Slepian-based encoder requires manual specification of the target region $R$ and its associated hyperparameters: the global and regional bandlimits $L_g$ and $L_r$, and the cap radius $\Theta$. These choices benefit from domain knowledge, and increasing $L_r$ can substantially raise build and train time without improving downstream performance. Meaningful high-resolution caps additionally require dense training observations inside $R$, since the leading well-concentrated modes must be learnable from local data. The choice of downstream neural network architecture

also influences performance non-uniformly across tasks (Table 5), and we find no single best-performing architecture in our experiments. For very small regions, the Shannon number $N$ remains low even at high bandlimits, capping representational capacity at fine spatial scales such as block-level urban patterns or individual-object detection. Finally, while spherical, localized representations are essential in regions where Euclidean encodings break down, most notably at the poles and along the antimeridian, the gains over a well-chosen Euclidean encoder are smaller in mid-latitude regions where map projection distortions are mild.

**Conclusion.** Our results demonstrate that spatio-spectral concentration is a powerful inductive bias for geographic machine learning. Slepian encodings consistently outperform baselines, with gains driven by where representational capacity is allocated rather than embedding dimensionality alone. More notably, our hybrid architecture shows that global geographic context improves local prediction, and concentrated local context improves global tasks such as species distribution modeling. This finding challenges the conventional separation between local and global modeling paradigms in spatial representation learning (Rolf et al., 2024). Combined with strong computational efficiency and natural extension to temporal encoding via DPSS, Slepian functions offer a unified framework for high-resolution, region-aware representation learning. Our framework naturally extends to adaptive-resolution geographic location encoders that learn region boundaries or attention weights over multiple regional bases, removing the need for manual region and hyperparameter specification that our current approach requires. Additionally, contrastive pretraining in the style of SatCLIP (Klemmer et al., 2025a) could yield general-purpose embeddings with regional Slepian heads for transfer to localized downstream tasks.

## Impact Statement

Applications of machine learning models that require geographic context span a range of societally important applications, from species distribution modeling (Beery et al., 2021; Teng et al., 2023; Mickisch et al., 2025; Cole et al., 2023; Mac Aodha et al., 2019; Lange et al., 2025), crop mapping (Tseng et al., 2025a; 2023), climate modeling (Karimzadeh et al., 2025), to disaster mapping and humanitarian assistance (Gupta et al., 2019). Several of these application spaces span clustered regions on Earth and are not phenomena that span the entire globe. Furthermore, the computational efficiency of our method and effectiveness of our hybrid encoder even with low-expressivity neural networks enables its usage in compute-scarce environments.

Potential negative impacts stem from downstream uses of improved geospatial prediction, including surveillance, targeting, or other forms of harmful monitoring (Tuia et al.,

2025). We encourage usage of our hybrid Slepian encoder in collaboration with stakeholders and domain scientists who are better informed on mission-critical regions that demand high representational capacities, and effective resolutions required for a task.

## Acknowledgments

A majority of training runs conducted in this work were run on an NVIDIA Grace-Hopper (GH200) GPU node provided by the University of Colorado Boulder's high performance computing system Alpine. We thank Brandon Reyes and the RC computing team at CU Boulder for allowing access to this resource. Alpine is jointly funded by the University of Colorado Boulder, the University of Colorado Anschutz, Colorado State University, and the National Science Foundation (award 2201538). We also acknowledge funding by the Deutsche Forschungsgemeinschaft (DFG, German Research Foundation) – project number 572735710. We additionally thank Nico Lang and the anonymous reviewers for providing us with valuable feedback at various stages of this work.

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

*Table 3.* **Species distribution modeling results on SINR benchmark.** Mean Average Precision (mAP) on S&T Birds (535 species) and IUCN range maps (2418 species). LinNet uses a linear classifier directly on the position encoding; ResidualFCNet uses a deep network. Hybrid Slepian combines global SH ($L_g$=10) with regional Slepian functions ($L_r$). Slepian-only uses regional functions without global SH ($L_g$=0). We observe that high-degree SH exhibits numerical instability with LinNet.

| | | | LinNet | | ResidualFCNet | |
|---|---|---|---|---|---|---|
| Encoding | Config | Dim | S&T | IUCN | S&T | IUCN |
| `wrap()` (Mac Aodha et al., 2019) | — | 4 | 0.250 | 0.009 | 0.732 | 0.627 |
| | $L_g$=10 | 100 | 0.637 | 0.342 | **0.736** | 0.696 |
| SH (Rußwurm et al., 2024) | $L_g$=30 | 900 | 0.560 | 0.102 | 0.716 | 0.691 |
| | $L_g$=40 | 1600 | 0.440 | 0.069 | 0.673 | 0.645 |
| | $L_r$=10 | 16 | 0.264 | 0.011 | 0.730 | 0.679 |
| Slepian-only ($L_g = 0$) | $L_r$=20 | 50 | 0.269 | 0.011 | 0.711 | 0.682 |
| | $L_r$=30 | 101 | 0.271 | 0.013 | 0.692 | 0.665 |
| | $L_r$=40 | 167 | 0.298 | 0.013 | 0.678 | 0.656 |
| | $L_r$=10 | 116 | 0.695 | 0.433 | 0.711 | **0.704** |
| Hybrid Slepian ($L_g = 10$) | $L_r$=20 | 150 | **0.704** | 0.460 | 0.711 | 0.701 |
| | $L_r$=30 | 201 | 0.696 | 0.478 | 0.691 | 0.702 |
| | $L_r$=40 | 267 | 0.672 | **0.479** | 0.665 | 0.694 |

## A. Additional Quantitative Experiments

Here, we conduct additional experiments to (i) expand on the experimental setup and results of our multi-cap, hybrid Slepian-SH encoder on the presence-only species distribution modeling task with SINR, (ii) study the embedding attribution of our image + location embedding experiment on the building density regression task using GradientSHAP values, (iii) demonstrate the effectiveness of our hybrid Slepian encoder on a fundamentally global land-ocean classification task with masked Slepians, and (iv) extend our Slepian encoder to concentrate representational capacity in the *temporal* domain.

### A.1. Presence-only Species Distribution Modeling with our Hybrid Slepian Encoder

To test the effectiveness of the hybrid nature of our Slepian-spatial encoder, we propose utilizing a presence-only species distribution modeling task that requires both local and global context. We follow the experimental framework proposed in SINR (Cole et al., 2023). In SINR-style species distribution modeling, the supervision signal is fundamentally different from standard geographic regression and classification tasks used in Section 4. We do not have reliable absence labels. Instead, we only observe presence events: a species was recorded at a location, but the vast majority of (species, location) pairs are unknown. SINR trains a multi-label classifier over species by pairing each observed presence with pseudo-negatives that are generated on the fly. Concretely, for a presence sample $(x, s)$, SINR samples either (i) a random background location $x_{bg}$ and treats the species as absent there, or (ii) a random "background species" and treats it as absent at the observed location, or (iii) both, depending on the loss variant. This training setup means the location encoder must be queried not only at observed locations, but also at a large number of randomly sampled global locations produced at each step of training. We train using the SINR $\mathcal{L}_{AN}$-*full* objective as described in Cole et al. (2023). Note that $\mathcal{L}_{AN}$-full requires evaluating the location encoder for many globally distributed pseudo-negative locations in every minibatch.

**Why a *hybrid* Slepian encoder is required.** This presence-only training regime motivates a hybrid positional encoding. Purely regional encoders are poorly matched to $\mathcal{L}_{AN}$-full because pseudo-negatives are sampled globally: if the encoding is only meaningful in a small region, the model receives unstable or uninformative features for most background samples. Conversely, purely global low-frequency encoders provide stability everywhere but can lack the spatial resolution needed to model fine-scale variation in regions where species ranges change rapidly or where observations are dense. Our hybrid encoder resolves this tension by combining a smooth global backbone with additional localized degrees of freedom targeted to regions of interest (ROIs).

**Hybrid multi-cap Slepian positional encoding.** We replace SINR's sinusoidal lon-lat encoding with a hybrid spherical encoder that combines a smooth global SH backbone with multiple high-resolution cap-Slepian blocks. Our feature map is

$$\Phi_{\text{Hybrid}}(x) = \text{Concat}\Big[\Phi_{\text{SH}}(x; L_{\text{global}}), \Phi_{\text{Cap}}^{(1)}(x), \dots, \Phi_{\text{Cap}}^{(K_c)}(x)\Big], \tag{6}$$

where $\Phi_{\text{SH}}(x; L_{\text{global}})$ is a low-degree SH encoding defined everywhere on $\mathbb{S}^2$, and each $\Phi_{\text{Cap}}^{(k)}(x)$ is a cap-Slepian feature block tied to an ROI cap. In our experiments we use $K_c = 2$ caps: one centered over the continental United States with angular radius $25°$, and one centered over Europe with angular radius $20°$ (Figure 7). These cap blocks provide additional degrees of freedom that concentrate spatial resolution within their respective ROIs, while the SH backbone maintains a globally coherent representation required by presence-only pseudo-negative sampling.

To make training efficient under $\mathcal{L}_{\text{AN}}$-full (which requires encoding many globally sampled pseudo-negative locations per minibatch), we *precompute* the full hybrid feature map $\Phi_{\text{Hybrid}}(x)$ over a dense global longitude-latitude grid once and cache it as a raster. At training time, we describe each observed location $x$ and each pseudo-negative background location $x_{\text{bg}}$ by *interpolating* its feature vector from this cached raster. This grid-based precompute and interpolation avoids repeatedly evaluating a large number of spherical basis functions inside the training loop, while still injecting region-specific capacity via the multiple cap-Slepian blocks.

Similar to (Cole et al., 2023) we evaluate on the S&T (eBird Status & Trends) dataset that evaluates range prediction for 535 bird species concentrated in North America, and the IUCN benchmark that provides a more taxonomically and geographically diverse benchmark using expert-curated range polygons from the International Union for Conservation of Nature. This task covers 2,418 species spanning birds, mammals, reptiles, and amphibians across all continents. While a Slepian-only encoder without global context might perform reasonably well on the S&T benchmark due to the larger presence within the cap region (United States), intuitively, the Slepian-only model should underperform on the IUCN benchmark which contains species range polygons in other parts of the world.

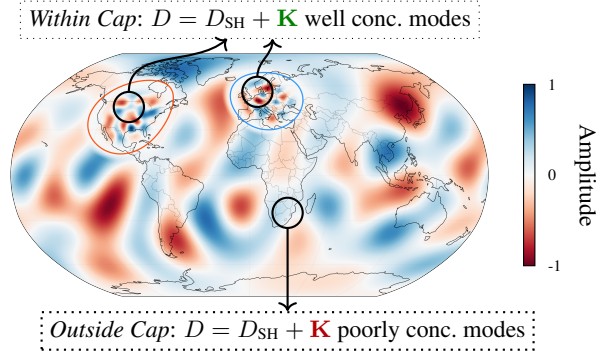

*Within Cap*: $D = D_{\text{SH}} + \mathbf{K}$ well conc. modes

*Outside Cap*: $D = D_{\text{SH}} + \mathbf{K}$ poorly conc. modes

*Figure 7.* **Training Spatial Implicit Neural Representations with our hybrid Slepian encoder.** Our two spherical Slepian caps span the United States and Europe due to the abundance of species observations in iNaturalist within these locations. Within-cap geographic locations receive well-concentrated, high-resolution embeddings from the Slepian component of the hybrid encoder. Outside-cap locations receive the same additional $K$ modes from our Slepian encoder. However, the Slepian component of these embeddings are poorly concentrated making the neural network rely on pure SH features.

**Results.** Table 3 presents species distribution modeling results across position encodings and classifier architectures. Our neural network architectures include a simple linear network (LinNet) and a deep residual network (ResidualFCNet) for location encoding, both proposed in (Mac Aodha et al., 2019).

Interestingly, we find that for a global species distribution modeling task, (i) Global spherical harmonics at higher resolutions *underperform* on both benchmarks regardless of the model architecture. While SH at $L=10$ achieves strong LinNet performance (0.637 S&T, 0.342 IUCN), increasing to $L=30$ or $L=40$ causes significant degradation (0.440 S&T, 0.069 IUCN at $L=40$). Even with ResidualFCNet, high-degree SH underperforms the $L=10$ baseline, suggesting that simply increasing SH bandwidth is not a viable path to finer spatial resolution.

Slepian-only encodings ($L_g=0$) perform poorly with LinNet across all regional bandwidths (0.26–0.30 S&T, 0.01 IUCN), comparable to the minimal `wrap()` baseline. This failure occurs because Slepian functions are spatially concentrated within their spherical caps—locations outside these regions receive near-zero representational capacity, making global species ranges linearly inseparable. However, increasing the expressivity with a larger ResidualFCNet compensates for this.

Combining global SH ($L_g=10$) with regional Slepian functions recovers strong LinNet performance (0.70 S&T, 0.48 IUCN at $L_r=20$–$30$) while achieving the best IUCN scores overall (0.704 with ResidualFCNet). The hybrid approach provides global coverage through the stable low-degree SH backbone while adding regional detail through numerically well-conditioned Slepian functions. Hybrid Slepian with 150 dimensions ($L_r=20$) outperforms SH with 900 dimensions ($L=30$) on both benchmarks with LinNet, demonstrating parameter efficiency. The IUCN benchmark shows the largest

*Table 4.* **Land-Ocean classification results with our Hybrid Slepian.** F1 score for spherical-harmonic (SH) and Hybrid Slepian encoders at two settings of the global bandlimit $L_g$, evaluated on three test-set regimes.

| Encoder | Coast. | Island | Unif. |
|---|---|---|---|
| SH ($L_g$=10) | 0.718 | 0.692 | 0.976 |
| Hybrid Slepian ($L_r$=10) | **0.725** | 0.693 | 0.975 |
| SH ($L_g$=30) | 0.652 | 0.646 | 0.966 |
| Hybrid Slepian ($L_r$=30) | **0.749** | **0.722** | **0.977** |

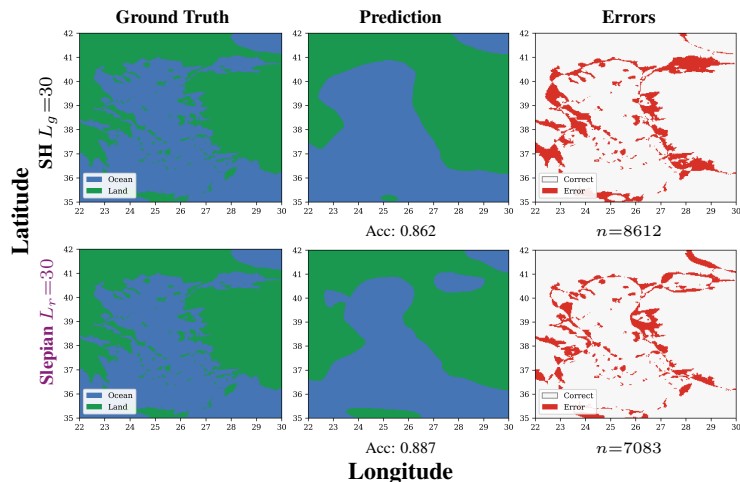

*Figure 8.* **Aegean Sea region comparison.** Ground truth land/ocean mask (left), model predictions (center), and error maps (right) for SH (top) and Hybrid Slepian (bottom) at $L_{g,r}$=30. We use $L_g = 10$ for the hybrid Slepian.

gains from hybrid encoding: LinNet IUCN improves from 0.342 (SH $L$=10) to 0.478 (Hybrid $L_r$=30), a 40% relative improvement. This suggests that the regional Slepian functions capture fine-grained spatial structure that benefits prediction of diverse taxa across varied geographies, where species ranges may have complex, localized boundaries.

### A.2. Slepian Encodings for global prediction tasks using an arbitrary mask

**Method.** Our Slepian encoder can also be extended to global tasks depending on the size and range of either the spherical cap or the arbitrary mask. To test the effectiveness of a masked Slepian encoder, we evaluate a hybrid Slepian-SH masked encoder on a synthetic land-ocean classification task (Rußwurm et al., 2024). The input to the location encoder is geographic location, and the output consists of a simple binary classification on whether the geographic location lies on landmass or ocean. A common failure case of geographic location encoders on the land-ocean classification task is the inability to accurately model fine-scale coastal boundaries or smaller, disjoint island chains. Recent work (Cai & Balestriero, 2025) showed this failure mode as the subgroup disparity in test cross-entropy loss between within-land and within-island regions.

To replicate the implementation of the unreleased `FAIR-Earth` package proposed in Cai & Balestriero (2025), we build a land-ocean dataset using the standard NaturalEarth 10m shapefile, along with the natural Earth 10m `minor islands` and `coastlines` shapefiles. The task is adapted to focus on high resolution challenging areas, i.e., islands and coastlines. In addition to a standard land-ocean dataset, we construct two separate test splits to measure performance on (i) islands, and (ii) coastlines. Land areas with area $< 30,000$ square miles are categorized as island, following (Cai & Balestriero, 2025). Our Slepian mask is a coarse coastline mask derived from Earth2014 topography data (Hirt & Rexer, 2015), with a buffer created using morphological operations. As in previous supervised geographic tasks we use a 3-layer MLP with ReLU activation as neural network.

**Results.** Table 4 and Figure 8 demonstrate that the Hybrid Slepian encoder provides modest improvements over the standard spherical harmonic (SH) baseline across coastline and island detection tasks, with more pronounced gains at higher bandlimits ($L_r$=30). The Slepian approach achieves better accuracy (0.887 vs. 0.862) and fewer misclassified pixels ($n$=7083 vs. $n$=8612) within high-frequency classification regions (for example, the Aegean Sea region visualized in Figure 8), particularly along complex coastlines where the SH basis exhibits characteristic smoothing artifacts. Table 4 also notes the deteriorating effect of resolution on Global SH's performance on challenging islands and coastlines, supporting our claim that increasing global resolution need not improve resolution-dependent performance on local regions. The hybrid Slepian-SH encoder benefits from an increase in resolution, especially on coastline and island boundaries. While statistically consistent, the improvements with masked Slepians remain marginal. Second, our experimental setup for the land-ocean masked Slepian task involves a degree of information leakage: the Slepian basis functions are constructed using a coarse coastline mask that shares structural similarity with the evaluation regions. Although the mask resolution differs substantially from the prediction targets, this circularity may partially inflate the reported gains. Future work should investigate the utility of Slepian representations for non-uniform, geometrically complex regions—such as archipelagos, fjords, or irregular

*Table 5.* **Comparison of positional encodings across neural network architectures.** We evaluate baseline encoders, spherical harmonics (SH), and our Slepian encoder on two geographic tasks. We use a standard *MLP* with ReLU activations, an MLP with residual connections and layer normalizations (*ResMLP*) (Touvron et al., 2023), and an MLP with learned feature gating (*GLU*) (Dauphin et al., 2017). For frequency-based encoders, we report best results from a resolution sweep. Best and second-best metrics per architecture are **bolded** and underlined. We omit results from Spherical Wavelets on the Arctic MSS dataset (🌀) due to the high computational cost on the large MSS dataset.

(a) California Housing ($R^2 \uparrow$)

| Method | Linear | MLP | ResMLP | GLU |
|---|---|---|---|---|
| Direct | 0.00±0.00 | -0.02±0.02 | -0.04±0.02 | 0.01±0.00 |
| Cartesian3D | 0.06±0.01 | 0.25±0.02 | 0.25±0.09 | 0.24±0.02 |
| Wrap | 0.05±0.02 | 0.25±0.02 | 0.25±0.06 | 0.22±0.05 |
| Grid | 0.24±0.00 | 0.45±0.02 | 0.55±0.06 | 0.60±0.04 |
| SphereC | 0.29±0.01 | 0.54±0.01 | 0.59±0.02 | 0.64±0.02 |
| SphereC+ | 0.29±0.02 | 0.44±0.07 | 0.54±0.05 | 0.61±0.03 |
| SphereM | 0.24±0.02 | 0.42±0.03 | 0.54±0.06 | 0.62±0.03 |
| SphereM+ | 0.24±0.01 | 0.42±0.06 | 0.53±0.04 | 0.61±0.03 |
| Theory | 0.25±0.01 | 0.42±0.03 | 0.58±0.02 | 0.62±0.02 |
| Wavelets | 0.23±0.01 | 0.34±0.01 | 0.35±0.02 | 0.34±0.02 |
| SH ($L=10$) | 0.24±0.02 | 0.48±0.02 | 0.57±0.02 | 0.65±0.02 |
| SH ($L=40$) | **0.32±0.01** | 0.64±0.02 | 0.69±0.01 | 0.71±0.01 |
| **Hybrid Slepian**($L=40$) | 0.30±0.01 | 0.63±0.01 | 0.69±0.01 | 0.71±0.01 |
| **Hybrid Slepian**($L=80$) | 0.27±0.01 | 0.68±0.01 | 0.71±0.01 | **0.76±0.01** |
| **Hybrid Slepian**($L=120$) | **0.32±0.00** | **0.69±0.01** | **0.71±0.00** | 0.74±0.01 |

(b) Japan Prefectures (Acc $\uparrow$)

| Method | Linear | MLP | ResMLP | GLU |
|---|---|---|---|---|
| Direct | 0.02±0.01 | 0.06±0.03 | 0.16±0.04 | 0.22±0.04 |
| Cartesian3D | 0.04±0.05 | 0.15±0.17 | 0.66±0.07 | 0.30±0.02 |
| Wrap | 0.05±0.05 | 0.23±0.14 | 0.66±0.07 | 0.32±0.01 |
| Grid | 0.64±0.02 | 0.82±0.01 | 0.79±0.05 | 0.83±0.02 |
| SphereC | 0.65±0.01 | 0.83±0.01 | 0.81±0.02 | 0.83±0.02 |
| SphereC+ | 0.66±0.01 | 0.83±0.00 | 0.82±0.02 | 0.84±0.01 |
| SphereM | 0.63±0.02 | 0.82±0.01 | 0.81±0.02 | 0.82±0.05 |
| SphereM+ | 0.64±0.01 | 0.82±0.01 | 0.80±0.01 | 0.82±0.01 |
| Theory | 0.66±0.01 | 0.84±0.00 | 0.82±0.01 | 0.84±0.01 |
| Wavelets | 0.51±0.04 | 0.80±0.01 | 0.80±0.04 | 0.81±0.04 |
| SH ($L=10$) | 0.59±0.02 | 0.82±0.01 | 0.81±0.02 | 0.84±0.02 |
| SH ($L=40$) | 0.82±0.01 | 0.87±0.00 | 0.86±0.01 | 0.87±0.01 |
| **Hybrid Slepian**($L=40$) | 0.82±0.01 | 0.87±0.01 | 0.85±0.01 | 0.87±0.01 |
| **Hybrid Slepian**($L=80$) | 0.86±0.00 | **0.89±0.00** | **0.88±0.01** | 0.89±0.01 |
| **Hybrid Slepian**($L=120$) | **0.87±0.00** | **0.89±0.00** | **0.88±0.01** | **0.90±0.00** |

(c) Arctic MSS Reconstruction (Score $\uparrow$)

| Method | Linear | MLP | ResMLP | GLU |
|---|---|---|---|---|
| Direct | 0.44±0.00 | 0.84±0.00 | 0.91±0.01 | 0.94±0.00 |
| Cartesian3D | 0.56±0.00 | 0.84±0.00 | 0.94±0.00 | 0.95±0.00 |
| Wrap | 0.65±0.00 | 0.89±0.01 | 0.95±0.00 | 0.96±0.00 |
| Grid | 0.68±0.03 | 0.87±0.01 | 0.88±0.06 | 0.95±0.01 |
| SphereC | 0.69±0.04 | 0.87±0.05 | 0.89±0.04 | 0.95±0.00 |
| SphereC+ | 0.71±0.04 | 0.86±0.06 | 0.88±0.04 | 0.95±0.01 |
| SphereM | 0.72±0.04 | 0.91±0.02 | 0.95±0.00 | 0.97±0.00 |
| SphereM+ | 0.73±0.04 | 0.90±0.03 | 0.95±0.00 | 0.96±0.00 |
| Theory | 0.72±0.06 | 0.88±0.06 | 0.88±0.03 | 0.76±0.03 |
| Wavelets | 🌀 | 🌀 | 🌀 | 🌀 |
| SH ($L=10$) | 0.77±0.00 | 0.86±0.01 | 0.92±0.00 | 0.97±0.00 |
| SH ($L=40$) | Diverge | Diverge | 0.85±0.00 | Diverge |
| **Hybrid Slepian**($L=40$) | 0.81±0.00 | 0.97±0.00 | 0.98±0.00 | **0.98±0.00** |
| **Hybrid Slepian**($L=80$) | 0.83±0.00 | **0.98±0.00** | **0.99±0.00** | 0.98±0.00 |
| **Hybrid Slepian**($L=120$) | **0.86±0.00** | **0.98±0.00** | **0.99±0.00** | 0.98±0.01 |

administrative boundaries—where spherical caps may be poorly suited and the benefits of spatially-concentrated basis functions can be more rigorously assessed.

### A.3. Feature Attribution Analysis

To understand the relative contribution of each positional encoding to the model's predictions, we train a unified model that jointly processes image embeddings, spherical harmonic (SH) coefficients, and Slepian function coefficients. The image embeddings from the AlphaEarth foundation model (Brown et al., 2025) (63 dimensions) are concatenated directly with learned projections of the SH coefficients (1600 dimensions for $L = 40$) and Slepian coefficients (6–12 dimensions depending on the region's Shannon number). Each positional encoding is processed by a separate two-layer MLP that projects it to a 256-dimensional representation before concatenation with the image features; the combined representation is passed through a final prediction head.

To quantify the importance of each input dimension, we employ GradientSHAP (Lundberg & Lee, 2017), a gradient-based attribution method that approximates Shapley values. For each test sample **x**, we estimate attributions by sampling $n$ reference points $\{\mathbf{b}_i\}_{i=1}^n$ from the training distribution, computing the gradient of the model output at a randomly

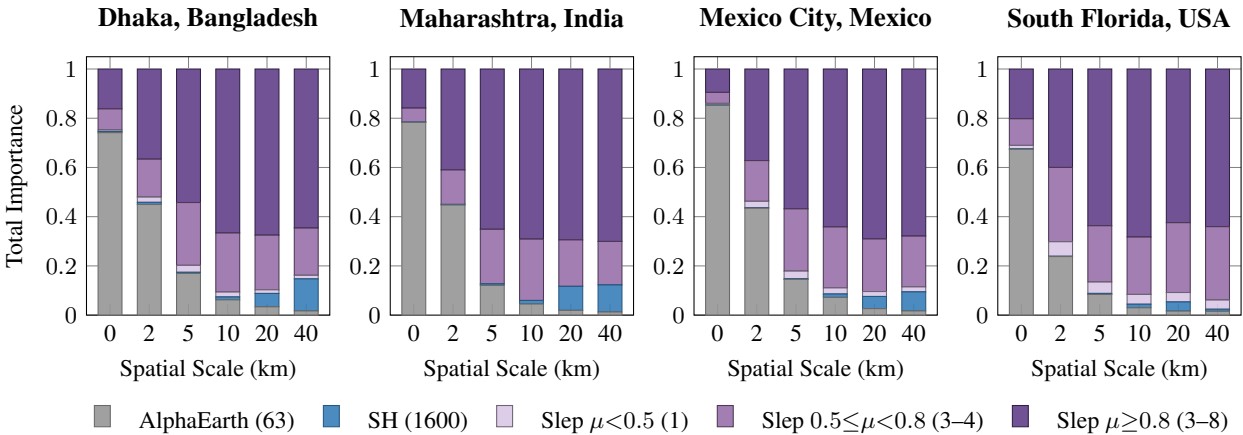

*Figure 9.* **Total location embedding attribution importance across spatial scales.** Stacked bars show the fraction of total model attribution (mean × dimensionality) for each embedding group, computed using GradientSHAP (Lundberg & Lee, 2017) averaged across 4 random seeds on a building density regression task (Sirko et al., 2021). Image embeddings from AlphaEarth (63-dim) dominate at fine scales (0 to 2km) but decline rapidly. At scales ≥5km, Slepian functions with $\mu \geq 0.8$ capture 55 to 70 % of total attribution despite comprising only 3 to 8 dimensions. Spherical harmonics (1600-dim) contribute minimally even at coarse scales.

interpolated point $\mathbf{b}_i + \alpha_i(\mathbf{x} - \mathbf{b}_i)$ where $\alpha_i \sim \text{Uniform}(0,1)$, and scaling by the input-reference difference:

$$\phi_j(\mathbf{x}) = \frac{1}{n} \sum_{i=1}^{n} \frac{\partial f(\mathbf{b}_i + \alpha_i(\mathbf{x} - \mathbf{b}_i))}{\partial x_j} \cdot (x_j - b_{i,j}). \tag{7}$$

We use $n = 25$ baseline samples per test point and report the mean absolute attribution $|\phi_j|$ to measure feature importance magnitude. Attributions are computed with respect to the raw input features (before the learned projections), allowing us to assess the contribution of each SH or Slepian coefficient directly. We aggregate attributions by embedding group—image, spherical harmonics, and our Slepian encoder binned by eigenvalue $\mu$—and report the *total importance*, which reflects each group's overall contribution to the prediction of building density value.

**Results.** Figure 9 shows a scale-dependent transition in feature importance that is consistent across all four regions. At fine scales (0–2 km), image embeddings dominate (67–85% of total attribution), but decay rapidly to $<2\%$ by 40 km. Well-concentrated embeddings from our Slepian encoder ($\mu \geq 0.8$) show the opposite trend: 10–20% at 0 km, growing to 54–70% at scales $\geq 5$ km—achieving this with only 3–8 features (depending on region). Spherical harmonics contribute minimally throughout ($<13\%$) despite their 1600 dimensions, supporting the hypothesis that global basis functions are inefficient for region-specific tasks. Among Slepian functions, eigenvalue concentration correlates with informativeness: the $\mu \geq 0.8$ band dominates, while the $\mu < 0.5$ band contributes negligibly, consistent with the theoretical interpretation that high-eigenvalue Slepian functions capture energy predominantly within the region of interest.

### A.4. Slepian-Based Positional Encoder for Spatio-Temporal Modeling

In Section 3.1, we introduced Slepian functions as spatio-spectrally concentrated solutions for regional geographic encoding. Many geospatial learning tasks, however, are inherently spatio-temporal: climate variables evolve over seasons (Watt-Meyer et al., 2023), species distributions shift with migration patterns (Mickisch et al., 2025), and land use changes over years (Tseng et al., 2023). A spatial encoder alone cannot capture these dynamics.

**Method.** We seek a temporal encoder $\Phi_{\text{Time}} : [-1, 1] \to \mathbb{R}^{K_t}$ that maps a normalized time coordinate $t$ to a feature vector, analogous to $\Phi_{\text{SH}}(x) \in \mathbb{R}^{D_L}$ for spatial coordinates. A natural starting point is the Legendre polynomial basis $\{P_k(t)\}_{k=0}^{K_t-1}$, which is orthogonal on $[-1, 1]$ and numerically stable. However, Legendre polynomials are not frequency-aware: they do not provide explicit control over which frequencies the representation can resolve. High-degree $P_k$ oscillate rapidly, but these oscillations follow polynomial structure rather than signal bandwidth, making the basis unable to preferentially represent signals with known spectral characteristics such as in the climate sciences. An alternative is the Fourier system $\{\sin(\pi k t), \cos(\pi k t)\}$, which directly encodes frequency content. Yet Fourier bases assume the signal is infinitely periodic, and real-world temporal data—such as a single year of climate observations—is a finite, non-periodic sequence. Truncating

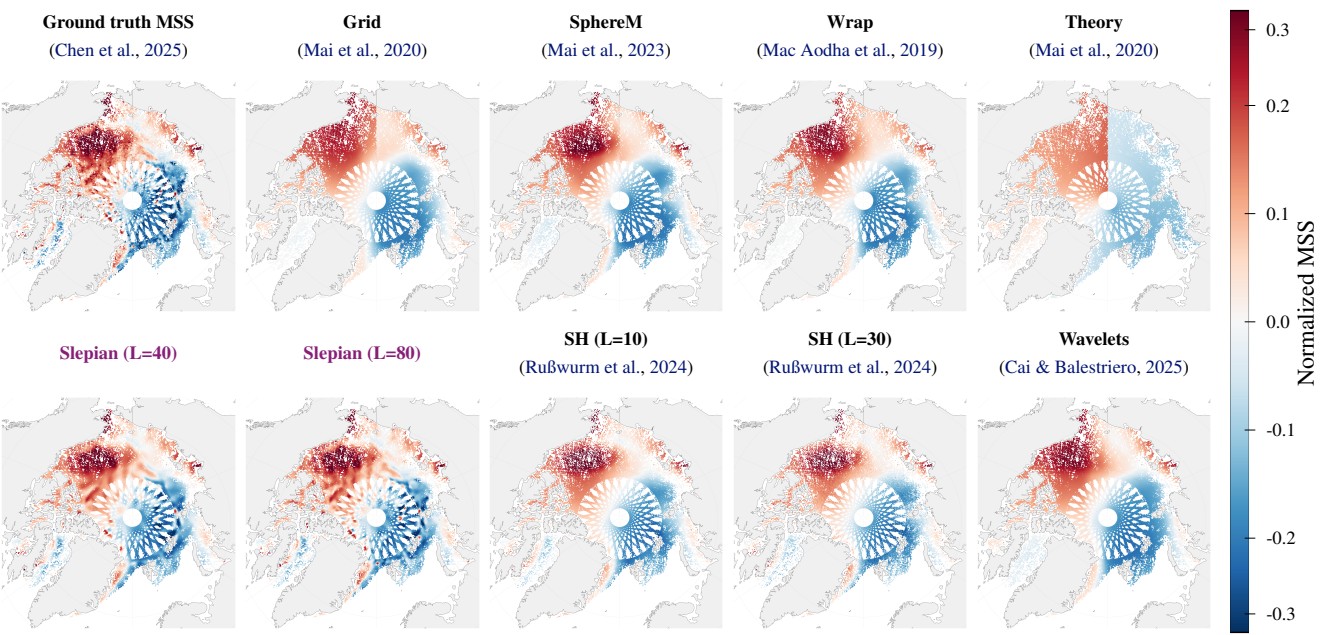

*Figure 10.* **Arctic MSS reconstruction results.** Ground truth MSS field (top left) and reconstructions from different location encoders. Slepian-based encoders (bottom left, purple labels) better preserve fine scale Arctic structure compared to global spherical harmonics and standard positional encoders. Condensed version in Figure 3.

a Fourier expansion to a finite window produces spectral leakage: the sharp edges of the window act as a rectangular taper, causing energy from each frequency to bleed into its neighbors and corrupting the representation (Harris, 1978).

Motivated by Slepian (1978)'s extension of spatial Slepian functions to discrete sequences, we introduce Slepians as a principled *temporal* basis that is concentrated in both time and frequency. Our Slepian-based temporal position encoder uses the Discrete Prolate Spheroidal Sequences (DPSS): a finite set of sequences optimally concentrated inside a prescribed frequency band $[-W, W]$, while being strictly supported on the finite observation window. This temporal encoder is the one-dimensional discrete-sequence analogue of the spatial encoder proposed in Equation (4). We seek sequences of length $N_t$ that maximize energy concentration inside a frequency band $[-W, W]$, where $W \in (0, \frac{1}{2})$ is the normalized half-bandwidth. The DPSS are the eigenvectors of a symmetric, positive-definite Toeplitz matrix $\mathbf{B} \in \mathbb{R}^{N_t \times N_t}$:

$$\mathbf{B}\,\mathbf{v}_k \;=\; \mu_k\,\mathbf{v}_k, \qquad B_{nm} \;=\; \begin{cases} 2W & n = m, \\ \dfrac{\sin\big(2\pi W(n-m)\big)}{\pi(n-m)} & n \neq m. \end{cases} \tag{8}$$

The eigenvalue $\mu_k \in [0, 1]$ measures the fraction of energy of $\mathbf{v}_k$ that lies within $[-W, W]$—the temporal analogue of the spatial concentration ratio $\mu$ in Equation (2). As in the spatial case, the eigenvalue spectrum exhibits a sharp transition: $\mu_k \approx 1$ for approximately $2N_tW$ leading sequences, and $\mu_k \approx 0$ thereafter. The quantity

$$K_t \;\approx\; 2N_tW \tag{9}$$

is the **temporal Shannon number**, counting the number of well-concentrated modes. This directly mirrors the spatial Shannon number $N(R, L_r) \approx \frac{\text{area}(R)}{4\pi}(L_r+1)^2$ from Equation (3): the time-bandwidth product $N_tW$ plays the role of the area-bandwidth product. We retain the leading $K_t = \lfloor 2N_tW \rfloor$ sequences to form our temporal basis.

The DPSS are defined on a discrete grid of $N_t$ points, but to allow our model to receive continuous time input $t \in [-1, 1]$, we evaluate each discrete sequence at arbitrary $t$ via cubic interpolation, yielding continuous functions $v_k(t)$ for $k = 1, \ldots, K_t$. We then apply a learnable linear projection $\mathbf{W}_t \in \mathbb{R}^{K_t \times K_t}$, initialized orthogonally, that mixes the interpolated DPSS features into the final temporal encoding:

$$\Phi_{\text{Time}}(t) \;=\; \mathbf{W}_t \left[\, v_1(t),\; v_2(t),\; \ldots,\; v_{K_t}(t) \,\right]^\top \;\in\; \mathbb{R}^{K_t}. \tag{10}$$

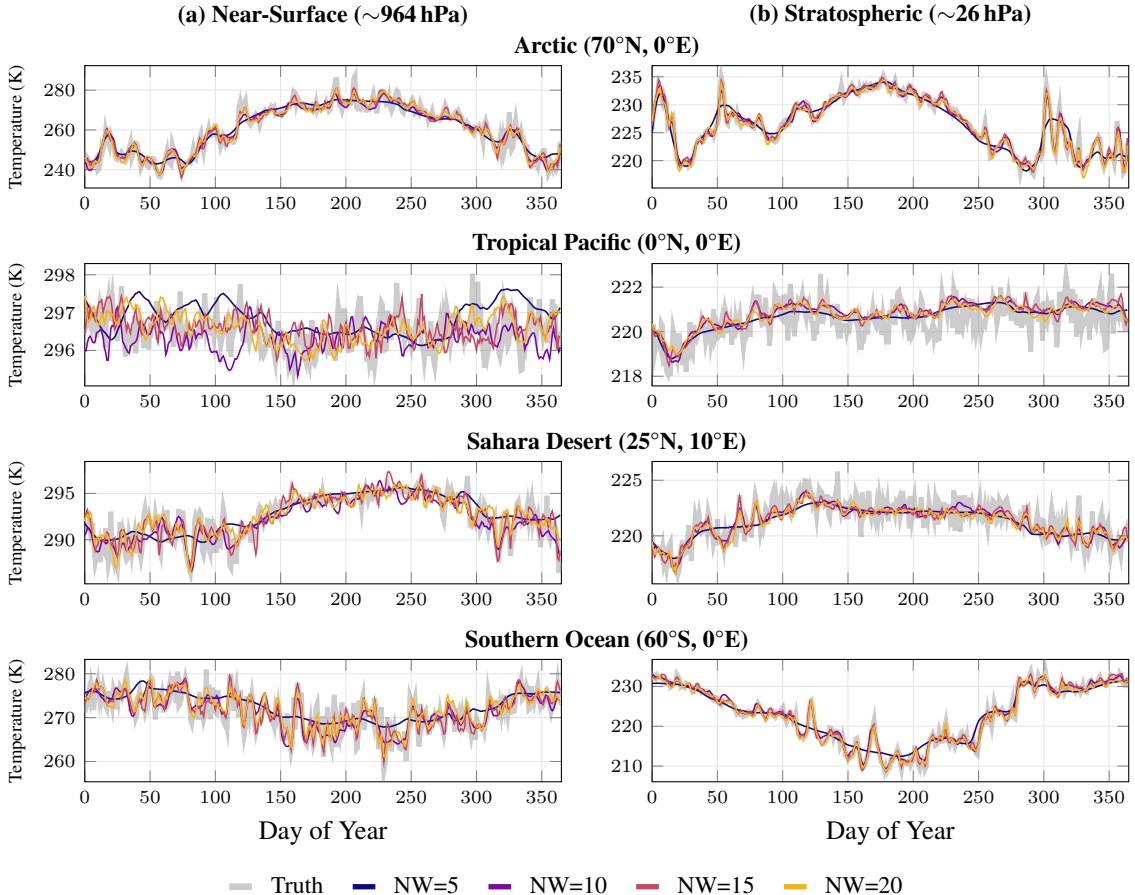

*Figure 11.* **Temporal reconstruction of two air temperature variables of the ACE dataset at four geographic locations using DPSS temporal encodings with varying bandwidth parameter** $NW$. (a) Near-surface temperature ($\sim$964 hPa). (b) Stratospheric temperature ($\sim$26 hPa). Lower bandwidth DPSS encodings underfit and are smooth reconstructions. Increasing $NW$ increases the regional temporal concentration, improving reconstruction quality.

This projection allows the network to learn task-specific linear combinations of the DPSS basis during training, while the underlying sequences remain fixed and provide a spectrally principled initialization. The half-bandwidth $W$ controls the trade-off between temporal resolution and frequency selectivity, analogous to how the cap radius $\Theta$ controls the spatial footprint in Section 3.2. A small $N_tW$ yields few, broadband sequences suited to slowly varying signals; a large $N_tW$ yields many, narrowband sequences that can resolve rapid temporal fluctuations.

We combine the spatial and temporal encoders into a **space-time encoder**, following Mickisch et al. (2025). Given a spatio-temporal coordinate $(x, t)$ with $x = (\lambda, \phi) \in \mathbb{S}^2$ and $t \in [-1, 1]$:

$$\Phi_{\mathrm{ST}}(x, t) = \mathrm{Concat}\big[\Phi_{\mathrm{SH}}(x), \Phi_{\mathrm{Time}}(t)\big], \tag{11}$$

which feeds into a neural network $f(x, t) = \mathrm{NN}\big(\Phi_{\mathrm{ST}}(x, t)\big)$. The spatial component captures geographic structure while the temporal component provides a spectrally principled, finite-window-aware representation that avoids the leakage of Fourier methods and the spectral agnosticism of polynomial bases.

**Dataset:** We evaluate a space-time encoder with a DPSS temporal basis on the AI2 Climate Emulator (ACE) dataset (Watt-Meyer et al., 2023), which provides output from a full-complexity atmospheric general circulation model. We use one year (2021) of simulation data, consisting of 12 monthly NetCDF files at $1° \times 1°$ spatial resolution ($360 \times 180 = 64{,}800$ grid points) and 6-hourly temporal resolution ($N_t = 1{,}460$ timesteps per year). Each spatio-temporal coordinate $(\lambda, \phi, t)$ is associated with 8 air temperature variables at distinct atmospheric pressure levels, yielding a regression target $y \in \mathbb{R}^8$. The full dataset comprises approximately 94.6M spatio-temporal points. We normalize longitude to $\lambda \in [-180°, 180°]$ and map the raw time coordinate linearly to $t \in [-1, 1]$. Each target variable is standardized to zero mean and unit variance

*Table 6.* **Temporal encoding comparison for atmospheric temperature prediction on ACE (Watt-Meyer et al., 2023).** We compare DPSS against polynomial (Monomial, Legendre), periodic (Fourier, Triangle), and ablation (No Time, Time Copy) baselines across eight pressure levels from stratosphere ($\sim$26hPa) to surface ($\sim$964hPa). RMSE (mean $\pm$ std across 3 seeds). **Bold**: best, underline: second best.

| | Stratosphere $\cdots\cdots\cdots\cdots\cdots\cdots\cdots\cdots\cdots\cdots\cdots\cdots\cdots\cdots\cdots\cdots\cdots\cdots\cdots\cdots\cdots\cdots\cdots\cdots\cdots\cdots\cdots\cdots$ > *Surface* | | | | | | | |
|---|---|---|---|---|---|---|---|---|
| **Model** | **T0 ($\sim$26hPa)** | **T1 ($\sim$99hPa)** | **T2 ($\sim$203hPa)** | **T3 ($\sim$337hPa)** | **T4 ($\sim$504hPa)** | **T5 ($\sim$690hPa)** | **T6 ($\sim$850hPa)** | **T7 ($\sim$964hPa)** | **Mean** |
| No Time | 6.230$\pm$0.000 | 6.074$\pm$0.001 | 4.989$\pm$0.000 | 4.357$\pm$0.001 | 5.342$\pm$0.002 | 5.612$\pm$0.002 | 6.020$\pm$0.001 | 7.003$\pm$0.004 | 5.703$\pm$0.001 |
| Time Copy | 1.326$\pm$0.043 | 1.942$\pm$0.041 | 2.404$\pm$0.019 | 2.142$\pm$0.008 | 2.798$\pm$0.009 | 2.873$\pm$0.013 | 2.987$\pm$0.013 | 2.951$\pm$0.012 | 2.428$\pm$0.018 |
| Triangle | 2.027$\pm$0.001 | 3.075$\pm$0.001 | 3.129$\pm$0.004 | 2.674$\pm$0.008 | 3.294$\pm$0.005 | 3.385$\pm$0.005 | 3.511$\pm$0.004 | 3.642$\pm$0.007 | 3.092$\pm$0.003 |
| Monomial | 1.013$\pm$0.105 | 1.562$\pm$0.152 | 2.027$\pm$0.181 | 1.819$\pm$0.150 | 2.350$\pm$0.215 | 2.389$\pm$0.226 | 2.535$\pm$0.221 | 2.572$\pm$0.179 | 2.033$\pm$0.178 |
| Legendre | 0.734$\pm$0.120 | 1.109$\pm$0.246 | 1.454$\pm$0.297 | 1.365$\pm$0.264 | 1.717$\pm$0.352 | 1.755$\pm$0.342 | 1.961$\pm$0.349 | 2.160$\pm$0.306 | 1.532$\pm$0.285 |
| Fourier | 0.717$\pm$0.050 | 1.054$\pm$0.103 | 1.407$\pm$0.132 | 1.319$\pm$0.122 | 1.642$\pm$0.145 | 1.684$\pm$0.147 | 1.895$\pm$0.166 | 2.101$\pm$0.155 | 1.477$\pm$0.128 |
| DPSS NW=15 | **0.657$\pm$0.012** | **0.951$\pm$0.008** | 1.278$\pm$0.015 | **1.190$\pm$0.011** | **1.490$\pm$0.009** | **1.532$\pm$0.014** | **1.723$\pm$0.017** | **1.935$\pm$0.013** | **1.344$\pm$0.012** |
| DPSS NW=17 | 0.703$\pm$0.014 | 1.044$\pm$0.010 | 1.354$\pm$0.016 | 1.288$\pm$0.012 | 1.595$\pm$0.009 | 1.638$\pm$0.015 | 1.857$\pm$0.011 | 2.061$\pm$0.013 | 1.443$\pm$0.012 |
| DPSS NW=19 | 0.667$\pm$0.009 | 0.963$\pm$0.014 | **1.269$\pm$0.011** | 1.206$\pm$0.016 | 1.498$\pm$0.012 | 1.540$\pm$0.008 | 1.739$\pm$0.010 | 1.977$\pm$0.018 | 1.357$\pm$0.011 |
| DPSS NW=25 | 0.680$\pm$0.015 | 0.994$\pm$0.011 | 1.303$\pm$0.009 | 1.254$\pm$0.013 | 1.534$\pm$0.016 | 1.585$\pm$0.012 | 1.810$\pm$0.014 | 2.037$\pm$0.008 | 1.400$\pm$0.010 |
| DPSS NW=35 | 0.688$\pm$0.011 | 0.994$\pm$0.017 | 1.288$\pm$0.013 | 1.254$\pm$0.009 | 1.527$\pm$0.011 | 1.583$\pm$0.015 | 1.816$\pm$0.012 | 2.061$\pm$0.016 | 1.401$\pm$0.013 |

using statistics computed on the training set. We randomly sample $1\%$ of all spatio-temporal points for each of the training, validation, and test sets ($\approx$ 946K points each), following the spatio-temporal interpolation protocol of Mickisch et al. (2025): the model must predict temperature at held-out locations and times drawn uniformly from the full spatio-temporal domain, rather than extrapolating to future timesteps. The task is thus $f: (\lambda, \phi, t) \mapsto y \in \mathbb{R}^8$, where $f = \mathrm{NN}\big(\Phi_{\mathrm{ST}}(\lambda, \phi, t)\big)$ and the loss is mean squared error on the standardized targets.

**Results.** Table 6 compares spatio-temporal position encoders for atmospheric temperature prediction across eight pressure levels on the ACE dataset. For all temporal encoders in Table 6, we use a global SH encoder with $L_g = 20$. DPSS encodings with bandwidth parameter NW=15 achieve the lowest mean RMSE (1.344), outperforming Fourier encodings (1.477) by approximately 9%. Performance exhibits a non-monotonic relationship with bandwidth: configurations in the range NW $\in$ [15,35] consistently outperform a Fourier temporal encoder, while extreme values degrade substantially. Notably, NW=5 (RMSE 2.536) performs worse than even raw time concatenation (2.428), indicating insufficient spectral resolution, whereas NW=50 (RMSE 2.251) suffers from over-smoothing despite utilizing $3\times$ more basis functions than the optimal configuration. This is confirmed through Figure 11. Low resolutions of DPSS temporal encoders produce coarse, smooth reconstructions of the temperature signal across several diverse geographic regions, while higher NW values increase the expressivity of the downstream neural network and produce higher-resolution reconstructions. This demonstrates that spectral concentration quality matters more than encoding dimensionality. The improvement is consistent across all atmospheric levels, suggesting that DPSS encodings effectively capture the characteristic temporal bandwidth of the atmospheric dynamics captured through the emulator.

## B. Implementation Details

### B.1. Positional Encoding Baselines

All encoders map geographic coordinates $(\mathrm{lon}, \mathrm{lat}) \in [-180, 180] \times [-90, 90]$ to a feature vector $\Phi(x) \in \mathbb{R}^d$. We organize these methods into four categories based on their underlying principles.

**Geometric Encodings.** The simplest geographic positional encoding approaches directly transform geographic coordinates into feature representations. The **Direct** encoder ($d = 2$) normalizes coordinates to the range $[-\pi, \pi]$ via $\Phi_{\mathrm{Direct}}(x) = (\pi \cdot \mathrm{lon}/180 - \pi, \pi \cdot \mathrm{lat}/180 - \pi)$. While computationally trivial, this representation suffers from the discontinuity at the antimeridian and fails to capture the spherical geometry of Earth. The **Cartesian3D** encoder ($d = 3$), used in Presto (Tseng et al., 2023) and other remote sensing foundation models, projects coordinates onto the unit sphere as $\Phi_{\mathrm{Cart3D}}(x) = (\cos \phi \cos \lambda, \cos \phi \sin \lambda, \sin \phi)$, where $\lambda$ and $\phi$ denote longitude and latitude in radians. This representation naturally handles the spherical topology but provides limited expressivity for complex spatial patterns. The **Wrap** encoder ($d = 4$), introduced by Mac Aodha et al. (2019) for presence-only species distribution modeling, applies trigonometric

wrapping to each coordinate independently: $\Phi_{\text{Wrap}}(x) = (\cos\lambda, \sin\lambda, \cos\phi, \sin\phi)$. This encoding handles the periodicity of longitude while providing a smooth representation suitable for gradient-based learning.

**Multi-Frequency Fourier Encodings.** Inspired by the success of positional encodings in neural radiance fields (Tancik et al., 2020) and the neuroscience of grid cells in mammalian navigation systems, Mai et al. (2020) proposed Space2Vec, a multi-scale representation learning framework. These methods apply sinusoidal functions at multiple frequency scales to capture spatial patterns across resolutions. Given $F$ frequency components, we employ geometric spacing where $f_i = 1/\tau_i$ with $\tau_i = \tau_{\min} \cdot \exp(i \cdot \Delta)$ and $\Delta = \log(\tau_{\max}/\tau_{\min})/(F-1)$. The **Grid** encoder ($d = 4F$) applies independent multi-frequency encoding to longitude and latitude: $\Phi_{\text{Grid}}(x) = \bigoplus_{i=1}^{F}[\sin(f_i\lambda), \cos(f_i\lambda), \sin(f_i\phi), \cos(f_i\phi)]$. The **Theory** encoder ($d = 6F$) extends this by projecting coordinates onto three unit vectors separated by $120°$, then applying the multi-frequency transformation to each projection. This design was motivated by the hexagonal firing patterns observed in grid cells (Mai et al., 2020). For all multi-frequency encoders, we use geometrically intialize the number of frequencies with $\tau_{\max} = 360$ and $\tau_{\min} \in [0.05°, 5°]$. We tune the $\tau_{\min}$ within this range in increments of $0.02°$.

**Sphere2Vec (Mai et al., 2023)** While Space2Vec encodings achieve multi-scale representation, they treat coordinates in a planar Euclidean space, leading to distortions—particularly near the poles—when applied to global geographic data. Mai et al. (2023) address this limitation with Sphere2Vec, a family of encodings based on the Double Fourier Sphere (DFS) representation that preserves spherical surface distances. The **Sphere**$^C$ encoder ($d = 3F$) constructs a multi-frequency 3D Cartesian embedding: $\Phi_{\text{Sphere}^C}(x) = \bigoplus_{i=1}^{F}[\sin(f_i\phi), \cos(f_i\phi)\cos(f_i\lambda), \cos(f_i\phi)\sin(f_i\lambda)]$. The authors prove that this encoding preserves the spherical geodesic distance between any two points, unlike planar encodings which introduce systematic errors that grow with latitude. We also evaluate **Sphere**$^{C+}$ ($d = 7F$), which concatenates Sphere$^C$ with Grid, **Sphere**$^M$ ($d = 5F$), a mixed-frequency variant coupling scaled and unscaled coordinates, and **Sphere**$^{M+}$ ($d = 9F$), which concatenates Sphere$^M$ with Grid. We use an identical resolution tuning procedure of $\tau_{\min}$ for Sphere2Vec models as described in the previous paragraph.

**Spherical Wavelets (Cai & Balestriero, 2025)** Standard implicit neural representations exhibit striking performance gaps on high-frequency signals such as coastlines and islands. Wavelets address this through multi-resolution encoding. Our implementation uses butterfly wavelets with $S = 3$ scales and $R = 75$ rotation samples distributed on the sphere via Fibonacci sampling, yielding $d = S \times R = 225$ features. Scale dilation follows $a_j = 2^{-(j+1)/6}$ for $j \in \{0, 1, 2\}$.

**Gaussian Process Baselines.** We implement several GP-based approaches to compare against kernel methods that have traditionally been used for spatial interpolation. We evaluate **Random Fourier Features** (RFF) (Tancik et al., 2020), which approximate shift-invariant kernels (RBF or Matérn) using random sinusoidal projections sampled from the kernel's spectral density, enabling linear-time training. Features are computed as $\sqrt{2/D}\cos(\mathbf{x}^\top\boldsymbol{\omega} + b)$ where $\boldsymbol{\omega}$ is drawn from the spectral distribution (Gaussian for RBF, Student-$t$ with df $= 2\nu$ for Matérn-$\nu$). Additionally, we implement **DeepRFF** (Chen et al., 2025), a multi-layer extension that stacks RFF layers with skip connections. Each layer consists of a frozen random projection (weights sampled from StudentT($2\nu, 1/\ell$)), a cosine activation scaled by $\sqrt{2\sigma^2/D}$, and a learnable linear output. Subsequent layers receive the concatenation of the previous layer's output and the original input coordinates.

### B.2. Neural Network Architectures

All positional encoders output a feature vector that is subsequently processed by a neural network head. We implement several architectures to ensure our findings are robust to this choice.

**Multi-Layer Perceptron.** Our default architecture is a 3-layer MLP defined as $\text{MLP}(z) = W_3 \cdot \sigma(W_2 \cdot \sigma(W_1 \cdot z + b_1) + b_2) + b_3$, where $\sigma(\cdot) = \text{ReLU}(\cdot)$ followed by dropout with probability $p = 0.1$. The hidden dimensions follow a tapering pattern $h \to h/2 \to 1$ (or $C$ for $C$-way classification), where $h = 128$ by default. This architecture provides sufficient capacity for learning non-linear mappings from location encodings to target variables while remaining computationally efficient.

**Residual MLP.** Following Touvron et al. (2023), we implement a residual architecture with residual blocks. Each block computes $h_{l+1} = h_l + \text{Linear}(\text{ReLU}(\text{Linear}(h_l)))$, with LayerNorm applied after the input projection. The residual connections facilitate gradient flow in deeper networks and have been shown to improve convergence in vision applications. Our default configuration uses hidden dimension 256, output dimension 128, and depth 4 blocks.

**Gated Linear Units.** The GLU architecture (Dauphin et al., 2017) employs a gating mechanism where each block computes $\text{GLU}(x) = \sigma(W_g x) \odot (W_v x)$, with $\sigma$ denoting the sigmoid function and $\odot$ element-wise multiplication. This gating allows the network to selectively propagate information, which can be beneficial when certain spatial features are more relevant than others. Our default configuration uses hidden dimension 256, output dimension 128, and depth 3 blocks.

### B.3. Training Configuration

We describe the training configuration for each experimental task. All experiments use the Adam optimizer unless otherwise specified, and employ early stopping based on validation loss to prevent overfitting.

**California Housing.** The California Housing dataset (Pace & Barry, 1997) comprises 20,640 census block groups from the 1990 U.S. Census, where the prediction target is median house value. We partition the data using a 60-20-20 train/validation/test split, consistent with prior work (Klemmer et al., 2025a). Target values are normalized to $[0, 1]$ via min-max scaling. Training proceeds for up to 200 epochs with learning rate $10^{-3}$, batch size 512, and early stopping patience of 20 epochs. To evaluate label efficiency, we train on random subsets comprising 1%, 10%, 25%, 50%, 75%, and 100% of the training data. Performance is measured using $R^2$ and mean absolute error (MAE) in dollars on the held-out test set, with results averaged over 5 random seeds. For the Slepian configuration, we compute basis functions for a $5°$ spherical cap centered on California $(37.0°\text{N}, 119.5°\text{W})$ with local bandlimit $L_{\text{local}} \in \{40, 80, 120\}$. The number of retained modes $K$ is determined by the eigenvalue threshold $\mu > 0.05$, which typically yields $K \approx L_{\text{local}}^2 \cdot A_{\text{cap}}/(4\pi)$ (the Shannon number).

**Japan Prefecture Classification.** To test fine-grained spatial boundary resolution, we construct a 47-class classification task using Japanese prefectures. We generate a labeled dataset by sampling points uniformly within each prefecture's administrative boundary, with the number of samples per prefecture varied from 1 to 100 to evaluate label efficiency. The data is partitioned using a 70-15-15 stratified split to ensure balanced class representation. Training uses cross-entropy loss with learning rate $10^{-3}$, batch size 256, and early stopping patience of 35 epochs, running for up to 200 epochs. We report top-1 accuracy on the held-out test set, averaged over 5 random seeds. Slepian functions are computed for a $10°$ cap centered on Japan $(36.0°\text{N}, 138.0°\text{E})$ with $L_{\text{local}}$ up to 120 and eigenvalue threshold $\mu > 0.05$ for mode selection.

**Arctic Mean Sea Surface.** The synthetic MSS dataset (Chen et al., 2025) provides approximately 1.2 million noisy observations generated by sampling a static mean sea surface field along realistic satellite ground tracks from CryoSat-2, Sentinel-3A, and Sentinel-3B. The task presents a challenging interpolation problem in the polar region, where standard latitude-longitude grids and many encoding schemes exhibit degraded performance. We use an 80-10-10 train/validation/test split with z-score normalization of target values. Training employs MSE loss with learning rate $10^{-3}$, batch size 2048, and early stopping patience of 30 epochs. To evaluate label efficiency, we train on fractions comprising 2%, 5%, 10%, and 100% of the training data. Performance metrics include $R^2$, RMSE (in meters), and MAE, averaged over 5 random seeds. For the Slepian configuration, we compute basis functions for a cap centered at the North Pole with radius matching the data extent (approximately $20°$), specifically testing the pole-safe property of Slepian functions.

**OpenBuildings Multi-Scale Regression.** For the geo-aware image regression task, we use the OpenBuildings dataset (Sirko et al., 2021) covering four diverse regions: Dhaka (Bangladesh), Mexico City (Mexico), Maharashtra (India), and Florida (United States). Each region is gridded, and for each cell we extract 63-dimensional embeddings from AlphaEarth (Brown et al., 2025) or 128-dimensional embeddings from Galileo (Tseng et al., 2025a). The prediction targets are log building density values, which we smooth using Gaussian filters at 9 spatial scales ($\sigma \in \{0, 1, 2, 3, 4, 5, 10, 20, 40\}$ km) to create a benchmark spanning high-frequency (block-level) to low-frequency (city-level) spatial patterns. The model architecture concatenates the image embedding with a location embedding projected through a 2-layer bottleneck (256 hidden units, ReLU activation), followed by a linear regression head. Training uses AdamW with learning rate $10^{-3}$ and weight decay $10^{-4}$, batch size 512, and early stopping patience of 20 epochs for up to 120 epochs. We report $R^2$, RMSE, and Moran's I of residuals to quantify remaining spatial autocorrelation. For each region, we define a spherical cap covering the study area with radius approximately 65% of the maximum spatial extent, computing Slepian bases at $L_{\text{local}} = 96$ with up to $K = 128$ modes retained. SH baselines use $L = 40$, representing the maximum tractable resolution for real-time computation.

## B.4. Slepian Function Computation

All Slepian function computations leverage the `pyshtools` library (Wieczorek & Meschede, 2018), which provides efficient implementations of the concentration eigenproblem for spherical caps. The computational workflow proceeds as follows: we first specify the spherical cap parameters (center longitude clon, center latitude clat, and angular radius $\theta$), then compute the Slepian eigenfunctions via `Slepian.from_cap(theta=..., lmax=...)`. The eigenfunctions are rotated from the pole to the desired cap center using `Slepian.rotate(clon=..., clat=...)`, and finally evaluated at query points via `to_shcoeffs(k).expand(lon=..., lat=...)`. For large-scale experiments such as OpenBuildings, we precompute Slepian design matrices $\Phi_{\text{Slep}}(x_i)$ for all grid points and cache them to disk, reducing per-experiment computational overhead from minutes to seconds.

## B.5. Hardware and Timing Methodology

All experiments in this work, including the efficiency runs reported in Figure 6, were conducted on a single NVIDIA Grace Hopper (GH200) compute node with 96 GB of HBM3 GPU memory. Wall-clock times are measured using Python's `time.perf_counter()`, with each measurement preceded by a warmup pass that is discarded before timing begins. To isolate compute cost from system noise, we pin thread counts for OpenMP, OpenBLAS, MKL, and NumExpr to one, evaluate all basis functions in single precision, and average over repeated runs.

The Build + Train Time reported on the $y$-axis of Figure 6 captures the full pipeline for each encoder. For Slepian encoders, basis construction includes solving the concentration eigenproblem on the spherical cap, rotating the basis to the target center, and evaluating each retained mode at all $N$ query coordinates. For SH baselines, basis construction reduces to analytically evaluating $(L+1)^2$ spherical harmonic functions at each coordinate. Both encoders feed into a downstream 3-layer MLP whose architecture is identical across configurations, with the number of input units set to the encoder's feature dimensionality: the Shannon number $K$ plus the coarse SH encoding (100 dims for $L_g = 10$) for Slepians and $(L+1)^2$ for SH. Table 7 summarizes the resulting trainable parameter counts. Because the Slepian feature dimensionality scales with the Shannon number rather than the full ambient SH dimension, the downstream MLP carries roughly an order of magnitude fewer trainable parameters at high bandlimits, which translates directly into the time and memory advantages observed in Figure 6.

*Table 7.* **Slepian encoders carry an order of magnitude fewer trainable parameters than matched SH at high bandlimits.** Feature dimension is the encoder's output size: $(L+1)^2$ for SH and the Shannon number $K$ for Slepians. Trainable parameter counts are for the downstream 3-layer MLP whose input width equals the feature dimension.

| Encoder | $L$ | Feature Dim | Trainable Params |
|---|---|---|---|
| SH | 10 | 100 | 21,249 |
| SH | 30 | 900 | 123,649 |
| SH | 40 | 1,600 | 213,249 |
| Slepian | 40 | 112 | 22,785 |
| Slepian | 80 | 144 | 26,881 |
| Slepian | 120 | 186 | 32,257 |

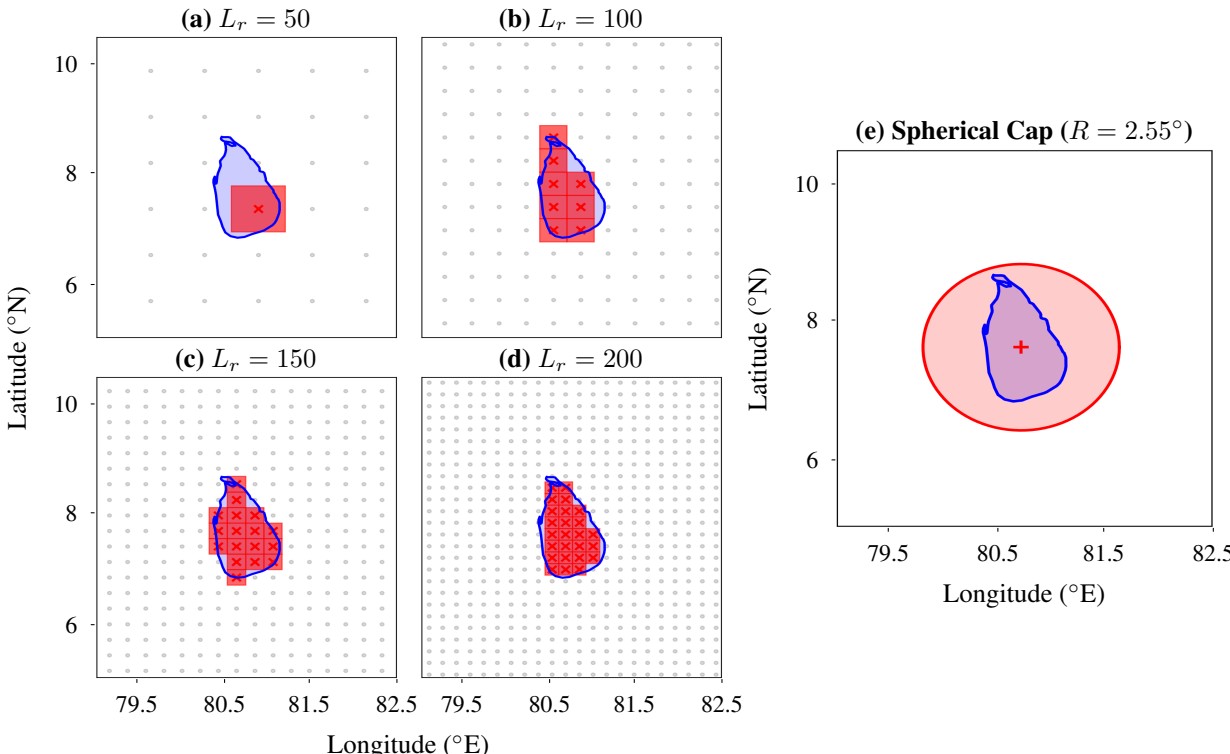

*Figure 12.* **Spherical caps yield accurate Slepian eigenvalues at any bandwidth $L$, while mask-based methods require high $L$ to resolve complex boundaries.** The bandwidth $L$ controls the sampling grid resolution: higher $L$ produces a finer grid with more points. Panels (a)–(d) show the *mask-based* approach for a small region (Sri Lanka, blue outline), where red cells mark grid points falling inside the region. At low $L$ (a), the coarse grid captures only a few points inside the boundary, distorting the eigenvalue spectrum and reducing the number of well-concentrated modes available for location encoding. Higher $L$ (b–d) improves boundary fidelity and recovers more usable modes, but grid size grows as $O(L^2)$, increasing computational cost. Panel (e) shows the spherical cap approach used in our work: a circle (red, $R = 2.55°$) centered on the region's centroid (+). Because caps have closed-form eigenvalue solutions, they yield the correct number of well-concentrated modes at any $L$ without expensive grid-based computation. However, this approach comes at the cost of including area outside the target boundary.

