# OpenReview forum: "Localized, High-resolution Geographic Representations with Slepian Functions"
_ICML.cc/2026/Conference — ICML 2026 regular_

### Official Review · Reviewer_hWHR · 2026-03-03

**Soundness:** 3
**Presentation:** 3
**Significance:** 3
**Originality:** 3
**Overall Recommendation:** 4
**Confidence:** 4

**Summary:**

The paper proposes a hybrid approach to encode geographic location that combines a spherical harmonics-based global basis and a Slepian function-based localized basis. The main contribution is combining a coarse global representation with fine-grained local representation to enable tradeoff between global and local context. For various localized downstream tasks (Cali Housing, Japan Prefectures, MSS), the proposed hybrid model outperforms the baseline encoding methods. The encoding method also outperforms other baselines in species distribution modeling with careful selection of spherical caps. The encoding method is also extended to model spatio-temporal geospatial data. The experiments on spatio-temporal modeling also show improvement over the baselines.

**Compliance With Llm Reviewing Policy:**

Affirmed.

**Final Justification:**

The proposed method has obvious limitations. However, I feel there is merit in accepting the paper and will lean towards acceptance. The authors should clearly discuss the limitations of their framework in the camera ready version.

**Key Questions For Authors:**

Please see Weaknesses.

**Limitations:**

Please see Weaknesses and try to address or discuss the limitations.

**Strengths And Weaknesses:**

**Strengths**:

1. The paper proposes Slepian functions to model functions on a spherical cap. This enables a location encoder to learn high resolution functions over a given spherical cap.
2. The paper is well written, well motivated and includes extensive experiments and discussions.

**Weaknesses**:

1. Theoretically, one can also change the default global coordinate projection system (e.g. EPSG:4326) to a more localized one to minimize errors for a given spherical cap. For instance, projection system EPSG:5070 is more suited for US than EPSG:4326 or EPSG:3857. How does changing the coordinate projection system itself perform against Slepian functions?

2. The biggest limitation of the framework is that it is **very sensitive to the spatial distribution of the training data itself**. It is overly embracing the spatial bias present in the datasets which are heavily sampled in USA and Europe. For instance, in species distribution modeling task (Figure 7), the authors sample multiple spherical caps for solely USA and Europe. This is counterintuitive since Amazon is one of the highly biodiverse regions on the Earth. This raises concerns about the framework for cases where there is a large domain shift between training and testing datasets.

3. Along point 2, what happens when the training and testing datasets are uniformly distributed across the globe? One of the limitations of the framework is that the region of interest need to be specified apriori as a spherical cap. How does the model adapt when no region of interest is specified and the entire globe is our region of interest? All the downstream tasks shown are locally concentrated. The paper would be complete if authors can include some global tasks such as Fibonacci-Lattice Checkerboard, land-ocean classification, ecoregion classification and biome classification.

4. For the Hybrid setting, why only consider SH for the global encoding (component B in Figure 2)? Can you also try and combine RFF or Space2Vec with Slepian instead of SH in the hybrid setting?

5. The authors talk about the problem of capturing high resolution representations over the Earth. What is sufficiently high resolution? Is it 1Km, 10Km or 10m? How do you measure the resolution of your learned representations? Can you compare the spatial resolution of embeddings across various location encoders?

6. Is there any relationship between classification error and Latitude/Longitude? Are there specific regions on the Earth where the model has high error? To this end, can you plot error maps related to the various downstream tasks?

7. The authors should cite and discuss a related work RANGE [1], which proposes a similar hybrid approach that combines SH with image-augmented embeddings. The authors should also cite and discuss HSR-SDM [2] that combines implicit and explicit neural representations for species distribution modeling.

**Justification for rating**:

The ideology/problem setting in itself is not novel since many papers in the past have proposed hybrid/multi-scale representations for various geospatial tasks. However, I do see the value in the approach using Slepian functions which have not been explored in the past. Currently, I am leaning towards very weak acceptance but would consider the comments from other reviewers and the author response to arrive at the final decision.

References

[1] Dhakal, Aayush, et al. "RANGE: Retrieval augmented neural fields for multi-resolution geo-embeddings." Proceedings of the IEEE/CVF Conference on Computer Vision and Pattern Recognition. 2025.

[2] Yuan, Shiran, and Hao Zhao. "Hybrid spatial representations for species distribution modeling." arXiv preprint arXiv:2410.10937 (2024).

---

> ### Author Rebuttal · Authors · 2026-03-30
>
> Thank you for your careful reading of our work and the constructive feedback.
>
> ### [W1] On utilizing an alternate co-ordinate projection system.
>
> We agree that using local projections are a practical heuristic for a localized setting. However, conic projections (EPSG:5070/EPSG:3995) sacrifice the sphere-native properties that our work and [1,2] explicitly preserve, namely pole-safety and spherical-surface distance. [2] discusses why these properties are essential for spatial representation learning. Slepian functions are defined on the sphere and do not produce distortions on $S^2$. Additionally, the multi-cap design (Section 3.3) would require stitching multiple local projections, risking cross-region interference. Lastly, changing the input coordinate system does not change where the basis functions allocate representational capacity or spectral energy. A Fourier/SH encoder on projected coordinates still allocates capacity uniformly over the domain.
>
> [1] Rußwurm et al., "Geographic Location Encoding with Spherical Harmonics and Sinusoidal Representation Networks," ICLR 2024
> [2] Mai et al., "Sphere2Vec: A General-Purpose Location Representation Learning over a Spherical Surface for Large-Scale Geospatial Predictions," ISPRS 2023
>
>
> ### [W2] On sensitivity to the spatial distribution of training data
>
> Meaningful high-resolution Slepian caps indeed require dense training data. We also investigated geographic domain shifts. As detailed in Section 4.1 and A.1, the IUCN benchmark contains mostly out-of-cap observations in Africa and South America [3]. We observe a 40% relative improvement on this domain-shifted test set compared to vanilla SH when using a Linear Network. High-resolution within-cap context (USA and EU) improves out-of-distribution performance.
>
> [3] Cole et al., "Spatial Implicit Neural Representations for Global-Scale Species Mapping," ICML 2023
>
> ### [W3] On performance of Slepians on global tasks
>
> For truly global tasks where data is uniformly distributed and resolution needs to be maintained over the entire globe, Spherical Harmonics are almost always the better choice. As detailed in Section 3.3 (L. 217), Slepian functions reduce to global SH when computed over $S^2$. However, many geospatial problems are also concentrated globally on land masses. For these cases Slepian functions with a masked region of interest are a reasonable choice even for a global task, as we tested in Appendix A2 (Tab 4, Fig 8).
>
> ### [W4] On considering alternate global encodings with Slepians.
>
> Slepian functions are finite linear combinations of SH. Because they share the same underlying basis ($S^2$), concatenating them to produce our hybrid representation is mathematically principled. Space2Vec and Planar RFFs are Euclidean encoders that suffer from the map projection distortions ([W1]). Combining Euclidean encodings for global context with local Slepian encodings would be an interesting direction, but would ultimately re-introduce these distortions (especially at the poles). Therefore, we decided to not include such experiments in our paper.
>
> ### [W5] On the spatial resolution of learned representations
>
> Quantifying the true resolution of continuous location encoders is an open area in spatial representation learning. The resolution of SH and Slepians is determined by the maximum spatial frequency the basis functions can resolve. For global SH, this is roughly 20,000/L km at degree L (L. 104). However, the optimal resolution depends on the task and the data sampling rate. When approximating the underlying geospatial field from individual data samples, the model resolution is an important hyperparameter that can smooth the representation. This is similar to the Nyquist sampling theorem in signal processing where artifacts (aliasing) occur when a signal is sampled at high rates. For example, in Table 2, increasing $L_r$ with a hybrid Slepian does not significantly increase test performance for species distribution modeling. We measure the resolution of our Slepian encoding empirically by demonstrating reconstruction of sharp gradients on the MSS interpolation task (Fig 4) and testing R$^2$ against multiple spatial scales in Fig 5. We will discuss this in Sec 6.
>
> ### [W6] On the relationship between lat/lon and classification error
>
> Yes, geospatial ML broadly has studied spatial properties of model errors, most recently with the geo-bias framework introduced in [4]. As demonstrated in Fig 3, Slepian functions concentrate representational capacity within the cap region, meaning that prediction errors are actively minimized within our targeted spatial boundaries compared to out-of-cap regions with larger errors. We agree with the suggestion, and will add spatial error maps to the Appendix.
>
> [4] Wu et al., "TorchSpatial: A Location Encoding Framework and Benchmark for Spatial Representation Learning," NeurIPS D&B 2024
>
> ### [W7]
> We agree with this suggestion, and have now cited both.

---

> > ### Author Rebuttal · Reviewer_hWHR · 2026-04-01
> >
> > My concerns are well addressed by the rebuttal.

---

### Official Review · Reviewer_kKMN · 2026-03-11

**Soundness:** 4
**Presentation:** 4
**Significance:** 4
**Originality:** 3
**Overall Recommendation:** 5
**Confidence:** 3

**Summary:**

This work develops a new geographic location encoder based on Slepian functions. The authors show how this basis can be constructed and how it is pole-safe. Through comprehensive experimental verification, the authors find that training ML methods using their encoder outperforms ML methods trained on other standard encoders.

**Compliance With Llm Reviewing Policy:**

Affirmed.

**Final Justification:**

I continue to view this submission positively and maintain my score of 5.

**Key Questions For Authors:**

I have no key questions for the authors

**Limitations:**

The authors do not discuss in much detail their approaches limitations (aside from the need to incorporate global information through the hybrid approach). I'm not entirely what other limitations there might be, but more discussion on where this approach may fail/may not outperform existing methods would be good.

**Strengths And Weaknesses:**

## Strengths:

1. This paper is very well written and easy to follow.

2. The figures in the paper are well made and clear to interpret.

3. The authors demonstrate clear advantages over existing encoders, across a range of tasks. The authors also show promising results on using their method for spatial-temporal data and explore how local vs global + local information can improve performance.

4. The paper is well motivated, with the limitations of existing methods clearly presented.

## Weaknesses:

I identified no major weaknesses of this work (note, that I am not an expert in this area)

## Minor points:
1. The authors mention pole-safety before they define what it is.

2. In their description of the species distribution modeling task, the authors mention "pseudo-negatives", but I do not think this was defined. Maybe this is obvious to those who are in the field, but I was not sure what that meant.

3. The authors reference Table 3 before Table 2, and Figure 4 before Figure 3. I know this might be due to space constraints and formatting, but if possible I think it makes most sense to organize numerically.

4. I have a neuroscience background and the trade-off between local vs global spatial information is often modeled as a combination of grid and place cells. Do the authors think there could be any connection between the Slepian functions and place cells?

---

> ### Author Rebuttal · Authors · 2026-03-30
>
> Thank you for the positive feedback and the valuable suggestions for improvement. We address the minor points and limitations raised in the review below:
>
> ## Defining pole-safety, referencing figures, and introducing pseudo-negatives.
>
> Thank you for pointing this out. We were indeed constrained by space and had to strategically place the initial figures and tables. In our revised version, we have:
>
> 1. Moved the definition of pole-safety from Section 3.3 to early in the introduction, and re-ordering Table 3,2 and Figure 2,4
> 2. Defining the pseudo-negative sampling strategy for training on species presence-only observations (introduced in [1]) in early-Section 4.1.
>
> [1] Cole et al., "Spatial Implicit Neural Representations for Global-Scale Species Mapping," ICML 2023
>
> ## On the connection between local/global geographic location encoders and grid/place cells
>
> The connection to concepts in neuroscience and possibly, biological navigation is indeed interesting\! Existing geographic location encoders like Space2Vec [2] explicitly utilize multi-scale periodic functions inspired by grid cells. Biological systems pair grid cells with place cells to encode highly localized, environment-specific contexts. Conceptually, the mathematical properties of our Slepian functions offer a compelling parallel to place cells, as they concentrate high-frequency representational capacity within strict spatial or temporal boundaries.
>
> [2] Mai, Gengchen, et al. "Multi-scale representation learning for spatial feature distributions using grid cells," ICLR 2020
>
> ## On the discussion of limitations
>
> We indeed wished for a broader discussion of our hybrid encoder’s limitations, but were constrained by space. Our camera-ready version will contain the following discussion of limitations
>
> 1. Our approach requires manual specification of the target region $R$ and associated hyperparameters ($L_r$, $L_g$, cap radius $\Theta$), which may require domain knowledge for optimal performance. For example, increasing $L_r$ may significantly increase build+train time without improving performance.
> 2. As we demonstrate in Table 5 (Appendix A2), the choice of neural network architecture used in conjunction with the Slepian PE can significantly impact performance, and there is no consistent best-performing architecture across tasks.
> 3. Additionally, for very small regions, the Shannon number $N(R, L_r)$ remains small even at high bandlimits, limiting representational capacity for ultra-high resolution tasks such as detecting individual objects or block-level spatial patterns.
> 4. While spherical, localized representations are extremely important for some areas (especially the poles), but might not be required for areas where the divergence from Euclidean encodings is minimal (e.g. areas around the poles).

---

> > ### Author Rebuttal · Reviewer_kKMN · 2026-04-01
> >
> > All my minor concerns/comments were addressed. I thank the authors for their response!

---

### Official Review · Reviewer_uGjc · 2026-03-12

**Soundness:** 3
**Presentation:** 3
**Significance:** 3
**Originality:** 3
**Overall Recommendation:** 4
**Confidence:** 1

**Summary:**

In this paper, the authors introduce a geographic location encoder with spherical Slepian functions to bridge the trace off between global and local performance. The proposed approach is evaluated on several benchmarks regarding classification, regression, prediction, etc. The authors further show the that the learned representation achieves a good trade off between global and local performance.

**Compliance With Llm Reviewing Policy:**

Affirmed.

**Final Justification:**

Keep the same rating as the concerns are addressed.

**Key Questions For Authors:**

Please see the three main clarification questions in the weakness section.

**Limitations:**

Yes

**Strengths And Weaknesses:**

Strengths:
1. The presentation of the work is clear with detailed text explanation, tables and well designed visualizations.
2. The evaluation of the proposed method is clear and comprehensive, and authors did a great job justifying the claims through supporting data points.

Weaknesses:
1. For people not familiar with this topic, is it possible to derive a theoretical bound for table 1 by setting L to infinity? That seems to give the upper bound of these evaluation metrics.
2. Can authors further explain why for some evals, like S&T under LinNet, it is significantly improved at L = 10 comparing Slepian vs hybrid Slepian? While for ResFCNet case, the improvement is actually much less.
3. When authors claims Slepian functions are more efficient, can the authors add more details regarding the hardware and condition used to arrive at the runtime shown in the paper?

---

> ### Author Rebuttal · Authors · 2026-03-30
>
> Thank you for engaging with our work and the constructive feedback.
> Responses to Weaknesses
> ## [W1] Is it possible to derive a theoretical bound for Table 1 by setting $L$ to infinity?
>
> We thank the reviewer for this insightful question. Theoretically, an infinite-bandwidth spherical basis can approximate any square-integrable function on $\mathbb{S}^2$. However, with finite training data, increasing $L$ increases the ambient dimensionality of the encoding — scaling as $O(L^2)$ globally or $O(f_R \cdot L^2)$ regionally for Slepians, where $f_R = \text{area}(R) / 4\pi$ is the area fraction of the region-of-interest. Here, we point to results in [1] which show that the intrinsic dimension estimates of current Earth representations are an order of magnitude under their ambient embedding size, and that current encodings are already “too large” and unable to fully use their representational capacity. Additionally, current shallow neural networks struggle to fully utilize this vast representational capacity, meaning an infinite spatial basis would not yield proportional performance gains. Establishing a formal bound on prediction error as a function of $(L,\; N_{\text{train}},\; \text{smoothness of target})$ would require connecting spherical approximation theory with the learning dynamics of the downstream neural network. For instance, extending the NTK analysis of Fourier feature mappings in [2] to spherical bases. We believe this is a valuable direction for future theoretical work.
>
> [1] Rao et al., Measuring the Intrinsic Dimension of Earth Representations, ICLR 2026.
> [2] Tancik et al., Fourier Features Let Networks Learn High Frequency Functions in Low Dimensional Domains, NeurIPS 2020.
>
> ## [W2] On the improvement of S&T under LinNet vs ResFCNet with Slepian vs hybrid Slepian
> This is a great insight. This phenomenon illustrates the distinction between theoretical resolution and *effective resolution* that we discussed in our response to hWHR. Specifically, we believe that the effective resolution is a joint property of the location encoder, the downstream NN's capacity, and the task characteristics. At a coarse $L_r=10$ resolution, we find that the increased performance of a hybrid vs Slepian-only encoder can be attributed to the expressiveness of each NN architecture. A linear network cannot increase the resolution of its inputs, and the standalone Slepian encodings are too coarse for LinNet to form complex, fine-grained classifications. When injected with the global context an SH encoding provides, the linear model is augmented with a rich set of complementary spatial features. That is, in the linear setting, a richer positional encoding basis has a more direct effect on task performance. Conversely, a deep residual network like ResFCNet possesses the capacity to synthesize a coarse basis into complex decision boundaries. Thus, the benefit of adding these global SH features in our hybrid encoding is diminished in the local evaluation setting of S&T. Note that, however, in the out-of-distribution IUCN setting as shown in Table 2, this global SH component of our hybrid encoding does indeed improve on the Slepian-only component with a ResFCNet. We will expand on this result given the additional page allowed to authors.
>
> ## [W3] On greater detail surrounding hardware used to generate our Slepian efficiency claims.
> We agree with this suggestion, and will include an appendix paragraph under Section B detailing the hardware used and specifics surrounding build and train times detailed in Fig 6 of our paper. All experiments, including efficiency runs were conducted on a NVIDIA Grace-Hopper (GH200) compute node with 96 GB HBM3 GPU memory. The "Build + Train Time" on the y-axis of Fig 6 is wall-clock time measured with Python's `time.time()`, encompassing the full pipeline for each encoder: (1) feature construction (basis computation for Slepian, or SH evaluation during forward passes), and (2) model training with MLPs containing an identical architecture outside of the number of input units. We vary the number of input units to accommodate the different embedding sizes of SH and our hybrid Slepian encoding. The concentration property of Slepians where the Shannon number dictates the number of well-concentrated modes is particularly beneficial for efficiency.
>
>  Encoder | L   | Feature Dim | Trainable Params
> ---------|-----|-------------|------------------
> | SH      | 10  | 100         | 21,249
> | SH      | 30  | 900         | 123,649
> | SH      | 40  | 1,600       | 213,249
> | Slepian | 40  | 112         | 22,785
> | Slepian | 80  | 144         | 26,881
> | Slepian | 120 | 186         | 32,257
>
> The table above shows that vanilla SH, on account of the $O(L^2)$ relationship between resolution and embedding size, needs an order of magnitude more trainable parameters, compared to our Slepian encoder. The effects of this benefit are reflected directly in Fig 6.

---

> > ### Author Rebuttal · Reviewer_uGjc · 2026-04-02
> >
> > I thank the authors for the detailed explanation. Would keep my score of weak accept.

---

### Official Review · Reviewer_ppYY · 2026-03-13

**Soundness:** 3
**Presentation:** 3
**Significance:** 3
**Originality:** 3
**Overall Recommendation:** 5
**Confidence:** 5

**Summary:**

This paper addresses an important problem in geographical representation learning -- memory-efficient and numerically stable high-resolution representation. This problem is in particular severe in spherical harmonics (SH) based encoders. The proposed method solves a  Slepian concentration problem to find the SH components whose densities concentrate in the given regions of interest and uses these components instead of the full spectrum. It effectively reduces the computational costs of SH representations.

**Compliance With Llm Reviewing Policy:**

Affirmed.

**Key Questions For Authors:**

1. From how the paper describes it, it seems that the Slepian caps need to be pre-defined. What if the test data lie in a different region than the training data?

2. Can you give more justification on the selection of evaluation datasets? Most of them are not seen in previous literature. If it is because those datasets are more "global", can you do some experiments and show if your method still works in that case?

**Limitations:**

See "Key Questions For Authors".

**Strengths And Weaknesses:**

**Strengths**

*Soundness*: The problem setup, concept definitions, mathematical derivations, and experimental results are well-organized and seem correct to the best of my knowledge. Extensive ablation studies strongly support the claims.

*Presentation*: The paper is very well written. The motivation, definitions and related math backgrounds are smoothly introduced and very easy to follow. The visual illustrations help understanding the strengths, e.g. the pole-safety and in-cap/out-of-cap resolution differences.

*Significance*: Though the method is based on spherical harmonics location encoding, it sheds some light on the more general problem of memory-efficient high-resolution representation learning on spheres.

*Originality*: To the best of my knowledge, the proposed method is novel.

**Weaknesses**:

*Significance*:

1. The proposed method is evaluated on tasks and datasets that are relatively less commonly used and benchmarked. The most relevant use-case of SH representations is Earth observations (e.g. CO2, climate, temperature, remote sensing images). It is preferred if the authors can report results on these downstream tasks.

2. The quadratic dimension explosion and numerical instability problem is mostly tied to spherical harmonics representations. A imminent question is whether we should try to mitigate this problem, or just look for new representation methods other than spherical harmonics which do not have quadratic dimensions and numerical instability in the first place.

---

> ### Author Rebuttal · Authors · 2026-03-30
>
> Thank you for the accurate summary of our work and the positive feedback.
>
> # Responses to weaknesses
>
> ## [W1] The proposed method is evaluated on tasks and datasets that are relatively less common.
>
> Our evaluated tasks are established benchmarks in geographic location encoding. Our species distribution modeling tasks (S&T, IUCN) use exact datasets and protocols from [1-3], and recent advancements like RANGE [4]. California housing is widely benchmarked in general ML and SatCLIP [5]. The OpenBuildings building density task follows the geo-aware image prediction protocol of TorchSpatial [6]. Our ACE spatio-temporal task follows the evaluation setting proposed in the space-time encoder in [10]. Three of our five evaluation settings directly involve Earth observation data: the Arctic MSS task uses CryoSat-2 satellite altimetry measurements [7], the OpenBuildings task uses high-resolution satellite imagery from [8], and the ACE spatio-temporal task predicts atmospheric temperature from a full-complexity atmospheric general circulation model [9,10]. While global variables like climate and CO2 are certainly important benchmarks, these variables are predominantly smooth, low-frequency, and globally continuous phenomena. Vanilla SH are already well suited for these tasks. Our Slepian encoding is explicitly designed for high-frequency, localized spatial phenomena.
>
>
> [1] Mac Aodha et al., "Presence-Only Geographical Priors for Fine-Grained Image Classification," ICCV 2019 [2] Cole et al., "Spatial Implicit Neural Representations for Global-Scale Species Mapping," ICML 2023 [3] Mai et al., "Sphere2Vec: A General-Purpose Location Representation Learning over a Spherical Surface," ISPRS 2023 [4] Dhakal et al., "RANGE: Retrieval Augmented Neural Fields for Multi-Resolution Geo-Embeddings," CVPR 2025 [5] Klemmer et al., "SatCLIP: Global, General-Purpose Location Embeddings with Satellite Imagery," AAAI 2025 [6] Wu et al., "TorchSpatial: A Location Encoding Framework and Benchmark for Spatial Representation Learning," NeurIPS D&B 2024 [7] Chen et al., "Deep Random Features for Scalable Interpolation of Spatiotemporal Data," ICLR 2025 [8] Sirko et al., "Continental-Scale Building Detection from High Resolution Satellite Imagery," arXiv 2021 [9] Watt-Meyer et al., "ACE: A Fast, Skillful Learned Global Atmospheric Model for Climate Prediction," arXiv 2023 [10] Mickisch et al., "A Joint Space-Time Encoder for Geographic Time-Series Data," ICLRW 2025
>
> ## [W2] On moving beyond SH-based encodings due to the quadratic dimension explosion
>
> Thank you for this thoughtful question. The $O(L^2)$ scaling is not a unique flaw of SH, but rather a fundamental property of sampling continuous signals on a spherical manifold. While alternative Euclidean representations (e.g., planar grids or RFF) easily bypass this scaling, they do so at the cost of geometric integrity, reintroducing the map projection distortions and polar singularities that our work specifically avoids. Our Slepian framework represents a direct effort to mitigate this exact limitation while remaining natively on the sphere. By optimizing the basis for spatial concentration, Slepian functions retain the rigorous geometric guarantees of SH but break the global $O(L^2)$ bottleneck. For localized tasks, our method effectively reduces the required dimensionality to $O(\text{fractional area} \cdot L^2)$, offering a representation method that balances sphere-nativity with computational tractability.
>
> # Responses to Questions
>
> ## [Q1] What if the test data lie in a different region than the training data?
>
> The manual specification of Slepian caps is indeed a limitation of our work, and adaptively routing Slepian caps is ongoing work. However, our current hybrid representation is robust to the geographic OOD setting. We detail this evaluation setting in Sec 4.1 and Appendix Sec A.1, where we evaluate our Hybrid Slepian-SH representation on the IUCN benchmark, which is heavily populated with species observations in Africa and South America—regions completely outside our Slepian caps. As shown in Table 2, we observe a 40% relative improvement on this out-of-cap test set compared to a vanilla Spherical Harmonic baseline (when using a Linear Network). Our core takeaway here is that high-resolution local context within-cap aids the hybrid component of our encoding with test data in a different region. Local context helps global tasks. We have added a standalone limitations paragraph to Sec 6.
>
> ## [Q2] On selection of evaluation datasets and global datasets.
>
> We refer to our response to [W1] regarding the selection of evaluation datasets and their common use in prior work. Utilizing Slepian encodings for global tasks is viable. In Appendix Sec A.2, we evaluate our Slepian encodings computed on a global land-ocean classification task. From Table 4 and Fig 8, we observe modest performance improvements and higher fidelity segmentations on high-resolution islands and land-boundaries.

---

> > ### Author Rebuttal · Reviewer_ppYY · 2026-04-01
> >
> > Thanks. My questions are resolved.

---

### Decision · Program_Chairs · 2026-04-30

**Decision:**

Accept (regular)

**Comment:**

After the discussion phase, all reviewers recommended acceptance (2x Accept, 2x Weak Accept). They noted that the paper addresses an important problem, is well written, has novelty, and is robustly evaluated. The rebuttal addressed many reviewer concerns, for example by discussing the limitations of the proposed method in greater detail, clarifying certain evaluation details, and agreeing to improve the presentation in various ways (e.g., adding missing references). As a result, the AC decided to accept the paper. Please take the reviewer feedback into account when preparing the camera-ready version.